# Inhibiting the system $x_C^-$/glutathione axis selectively targets cancers with mutant-p53 accumulation

David S. Liu[1,2,3], Cuong P. Duong[2], Sue Haupt[1], Karen G. Montgomery[1], Colin M. House[1], Walid J. Azar[1], Helen B. Pearson[1,3,†], Oliver M. Fisher[4], Matthew Read[1,2,3], Glen R. Guerra[1,2,3], Ygal Haupt[1,3], Carleen Cullinane[1,3], Klas G. Wiman[5], Lars Abrahmsen[6], Wayne A. Phillips[1,2,3,7,*] & Nicholas J. Clemons[1,3,*]

*TP53*, a critical tumour suppressor gene, is mutated in over half of all cancers resulting in mutant-p53 protein accumulation and poor patient survival. Therapeutic strategies to target mutant-p53 cancers are urgently needed. We show that accumulated mutant-p53 protein suppresses the expression of *SLC7A11*, a component of the cystine/glutamate antiporter, system $x_C^-$, through binding to the master antioxidant transcription factor NRF2. This diminishes glutathione synthesis, rendering mutant-p53 tumours susceptible to oxidative damage. System $x_C^-$ inhibitors specifically exploit this vulnerability to preferentially kill cancer cells with stabilized mutant-p53 protein. Moreover, we demonstrate that *SLC7A11* expression is a novel and robust predictive biomarker for APR-246, a first-in-class mutant-p53 reactivator that also binds and depletes glutathione in tumours, triggering lipid peroxidative cell death. Importantly, system $x_C^-$ antagonism strongly synergizes with APR-246 to induce apoptosis in mutant-p53 tumours. We propose a new paradigm for targeting cancers that accumulate mutant-p53 protein by inhibiting the SLC7A11–glutathione axis.

[1] Division of Cancer Research, Peter MacCallum Cancer Centre, Melbourne, Victoria 3000, Australia. [2] Division of Cancer Surgery, Peter MacCallum Cancer Centre, Melbourne, Victoria 3000, Australia. [3] Sir Peter MacCallum Department of Oncology, The University of Melbourne, Parkville, Victoria 3010, Australia. [4] Gastroesophageal Cancer Program, St Vincent's Centre for Applied Medical Research, Darlinghurst, New South Wales 2010, Australia. [5] Department of Oncology-Pathology, Cancer Center Karolinska, Karolinska Institutet, Stockholm SE-171 76, Sweden. [6] Aprea Therapeutics, Nobels väg 3, Solna 171 65, Sweden. [7] Department of Surgery (St Vincent's Hospital), The University of Melbourne, Parkville, Victoria 3010, Australia. † Present address: Department of Biomedicine, European Cancer Stem Cell Research Centre, Cardiff University, Cardiff CF24 4HQ, United Kingdom. * These authors contributed equally to this work. Correspondence and requests for materials should be addressed to W.A.P. (email: wayne.phillips@petermac.org) or to N.J.C. (email: nicholas.clemons@petermac.org).

The tumour suppressor gene *TP53* is mutated in a large proportion of cancers. The loss of wild-type p53 (wt-p53) activity and acquisition of oncogenic gain-of-function, secondary to aberrant accumulation of mutant-p53 (mut-p53) protein, frequently results in aggressive tumour phenotypes and poor survival[1]. Therefore, effective therapies to target mut-p53 cancers are urgently needed.

APR-246 (PRIMA-1[met]) is the most clinically advanced mut-p53 targeting agent and has been shown to reactivate wt-p53 apoptotic functions[2]. This results in potent anti-tumour activity in preclinical models where drug sensitivity is strongly associated with levels of accumulated mut-p53 protein[3]. Recently, studies have shown that APR-246 can exert additional effects, particularly through antagonizing the glutathione (GSH) and thioredoxin reductase system[4,5], leading to increased reactive oxygen species (ROS). This fuels early speculation that there is potential cross-talk between mut-p53 and redox regulation[6].

Mounting evidence indicates that cancer cells produce higher levels of ROS compared to normal cells, which in turn can activate mitogenic signalling and promote carcinogenesis[7]. However, ROS can be a double-edged sword, as excessive accumulation leads to oxidative damage and cell death. These findings have led to the hypothesis that cancer cells with elevated ROS are sensitive to further oxidative insults and therefore can be selectively targeted. Despite compelling preclinical data, human trials of prooxidants have been disappointing[7]. Thus, it is critical to further elucidate the key modulators of redox balance to design strategies that maximally exploit the redox differential between normal and cancer cells.

In this study, we explore in detail the mechanisms and consequences of APR-246-induced oxidative stress. This led us to uncover a crucial link between mut-p53 and cellular redox modulation. We demonstrate that high levels of mut-p53, through binding to NRF2 and impairing its canonical antioxidant activities, directly promote ROS accumulation in cancer cells. This creates an inherent predisposition to further oxidative stress that can be therapeutically harnessed. APR-246 and inhibitors of the cystine/glutamate antiporter, system $x_C^-$, take advantage of this vulnerability to selectively kill mut-p53 cancer cells. In combination, these agents synergistically deplete mut-p53 cancers of GSH, leading to overwhelming ROS accumulation and extensive cell death. Importantly, we show that endogenous expression of *SLC7A11*, a key component of system $x_C^-$, reliably predicts tumour response to these therapies. Collectively, we propose a novel strategy to target cancers that accumulate mut-p53, based on perturbation of the SLC7A11–GSH axis.

## Results

### APR-246 triggers lipid peroxidation through depleting GSH.
Using oesophageal cancer as a model, we characterized the mechanisms and consequences of ROS induction by APR-246. Firstly, we verified across a panel of normal and cancer cell lines with differing p53 status (Supplementary Table 1) that treatment with a lethal dose of APR-246 resulted in higher intracellular ROS (Fig. 1a). Extending this, we confirmed that APR-246 rapidly decreases intracellular GSH in mut-p53 cells (Fig. 1b) leading to a concomitant increase in ROS levels, prior to the onset of apoptosis (Supplementary Fig. 1a,b). Furthermore, the pan-caspase inhibitor Q-VD prevented APR-246-induced apoptosis without inhibiting ROS elevation (Supplementary Fig. 1c,d). Altogether, these results suggest that ROS elevation precedes the activation of apoptotic pathways, and is not merely a consequence of cell death. Importantly, APR-246-induced ROS and apoptosis could be prevented by co-culturing with the antioxidants, *N*-acetyl-cysteine and GSH-monoethyl ester (Fig. 1c,d; Supplementary

Fig. 1e,f), highlighting ROS and the GSH pathway as key players in the anti-tumour activity of APR-246.

To determine how APR-246 depletes GSH, we applied mass spectrometry to show that 2-methylenequinuclidin-3-one (MQ), the active component of APR-246, reacts with GSH inside cancer cells (Fig. 1e,f). The resultant adduct (GS-MQ) has significantly reduced capacity to participate in GSH recycling via GSH reductase (Fig. 1g), a crucial step in maintaining GSH homeostasis. Given that intracellular GSH can detoxify xenobiotics through formation of GSH-drug conjugates[8], we compared residual GSH levels following APR-246 and chemotherapy treatments. At equi-efficacious doses, APR-246 depleted GSH to a greater extent than cisplatin, 5-fluorouracil, epirubicin, irinotecan and paclitaxel (Supplementary Fig. 1g), further emphasizing the importance of this phenomenon to the activity of APR-246.

We next investigated the consequences of APR-246-induced GSH depletion, and found that ROS accumulates in the mitochondria (Fig. 2a), a major reservoir for GSH, leading to lipid peroxidation (Fig. 2b; Supplementary Fig. 2a), mitochondrial rupture (Fig. 2c) and release of the apoptotic initiator, cytochrome *c* (Fig. 2d). Furthermore, using transmission electron microscopy, we observed a characteristic set of changes in the mitochondria after APR-246 treatment, beginning with organelle condensation and disrupted cristae architecture, followed by gross swelling, loss of outer membrane integrity and eventual rupture (Supplementary Fig. 2b). Importantly, the cytotoxic effects of APR-246 could be rescued with trolox, ferrostatin-1 and 2-mercaptoethanol (Fig. 2e), antioxidants that retard lipid peroxidation[9]. Incidentally, these are all potent inhibitors of ferroptosis, an iron-dependent, caspase independent form of cell death[9]. However, the iron-chelator deferoxamine (DFO) did not affect APR-246 activity (Supplementary Fig. 2c), suggesting that GSH depletion by APR-246 triggers lipid peroxidative, but not ferroptotic cell death.

### *SLC7A11* expression predicts tumour sensitivity to APR-246.
Having established that GSH depletion is an integral mechanism of APR-246 activity, we set out to identify predictive biomarkers using a targeted pharmacogenomics approach. On the basis of our finding that endogenous GSH levels correlated with APR-246 GI50 (Supplementary Table 1) in our cell line panel (Fig. 3a), we shortlisted genes involved in GSH synthesis and recycling, and correlated their mRNA expression with APR-246 sensitivity (Fig. 3b). Of these, *SLC7A11*, *SLC3A2* and *GPX1* were significantly associated with drug response. Notably, the two genes with the strongest correlation, *SLC7A11* and *SLC3A2*, together encode the cystine/glutamate antiporter system $x_C^-$, which serves to import cystine, the rate limiting substrate for the formation of GSH[10]. Since *SLC3A2* is dispensable for the function of system $x_C^-$ (ref. 11), we focused our attention on *SLC7A11* and found that while protein and mRNA expression varied widely between different cell lines (Fig. 3c,d), both correlated with resistance to APR-246 (Fig. 3e,f). Consistent with this, analysis of multiple microarray datasets from the NCI-60 cell line panel (a panel of 60 diverse human cancer cell lines used by the National Cancer Institute for therapeutic development[12,13]) demonstrated that *SLC7A11* expression correlated more strongly with response to the APR-246 lead compound PRIMA-1 than any other gene (Supplementary Fig. 3a,c). Remarkably, this relationship was evident across all of the nine different tumour types in the NCI-60 panel including both solid and haematological malignancies, highlighting the wide spread importance of this finding. Importantly, we noted that cell lines with missense *TP53* mutations, which are known to be more sensitive to APR-246 and PRIMA-1 (Fig. 3e; Supplementary Fig. 3d) also have lower

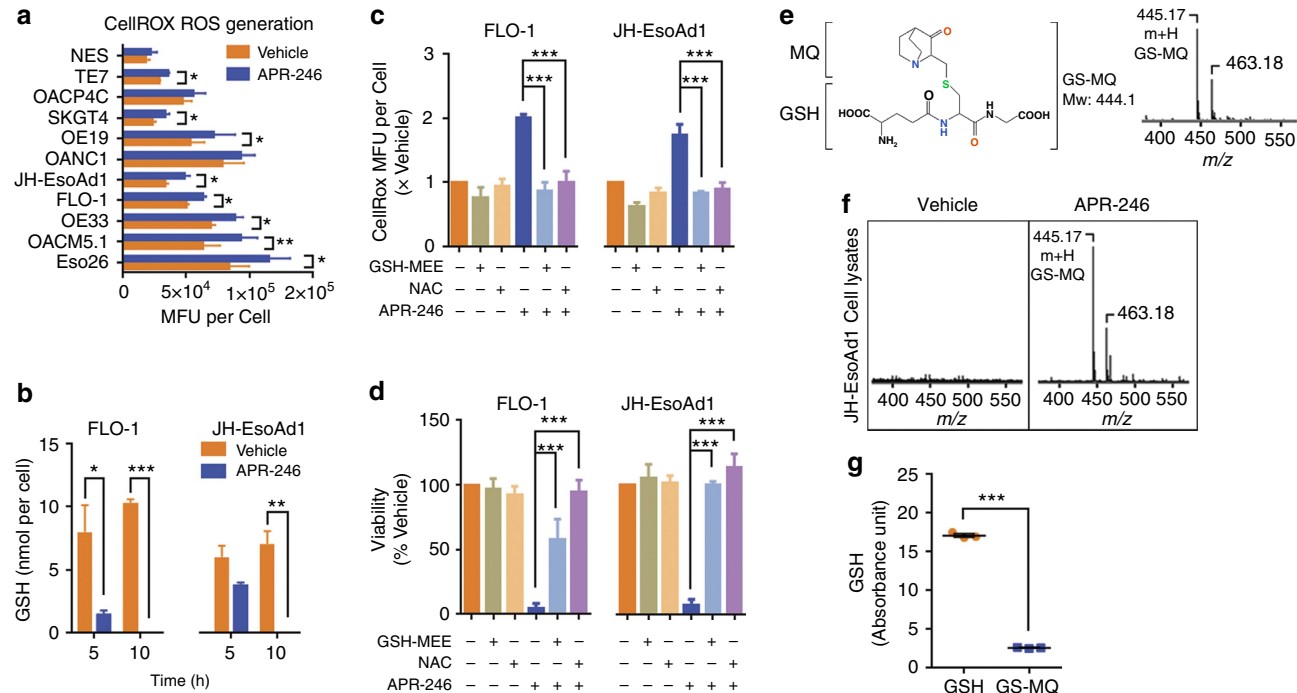

**Figure 1 | Glutathione depletion and ROS induction is central to the anti-tumour activity of APR-246.** (**a**) ROS detected using CellROX 6 h post 50 μM APR-246 treatment across a panel of oesophageal cell lines. Mean fluorescence unit (MFU) ± s.e.m. From here on FLO-1 and JH-EsoAd1 cells were treated with their GI90 doses: 25 and 40 μM of APR-246, respectively. (**b**) Glutathione (GSH) concentrations post APR-246 treatment in FLO-1 and JH-EsoAd1 cells. (**c,d**) CellROX (**c**) and viability (**d**) analysis post treatment with APR-246 and/or 5 mM glutathione-monoethyl ester (GSH-MEE) and/or *N*-acetyl-cysteine (NAC) in FLO-1 and JH-EsoAd1 cells assayed at 10 and 96 h, respectively. (**e**) Structural formula of GS-MQ (left) following reaction of MQ with GSH. Typical mass spectrometry (MS) pattern of 100 μM GS-MQ ($m/z = 445.17$ [m + H]) in 0.1% formic acid (right). (**f**) MS analysis of JH-EsoAd1 cell lysates collected 10 h post vehicle and APR-246 treatment. (**g**) Absorbance of 10 μM GSH or GS-MQ as measured with the Cayman GSH kit containing GSH reductase. Paired *t*-test (**a**), unpaired *t*-test (**b,g**), one-way ANOVA with Dunnett's multiple comparison post-test (**c,d**). Error bars = s.e.m., $n = 3$ for all, *$P < 0.05$, **$P < 0.01$, ***$P < 0.001$. See also Supplementary Fig. 1.

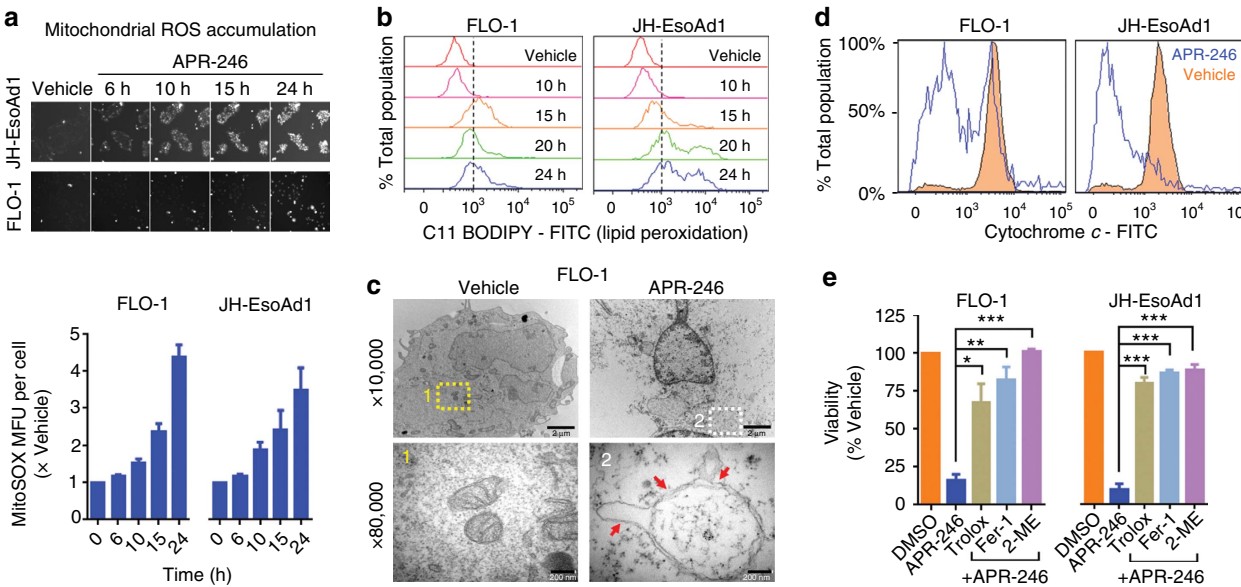

**Figure 2 | APR-246 triggers lipid peroxidative cell death through depleting glutathione.** (**a,b**) Detection of mitochondrial ROS using MitoSOX (**a**) and lipid peroxidation using C11-BODIPY (**b**) post APR-246 treatment in FLO-1 and JH-EsoAd1 cells. (**c**) Transmission electron microscopy of FLO-1 cells treated with APR-246 for 15 h. Red arrows: mitochondrial membrane rupture. A minimum of 10 cells were examined. Scale bar for ×10,000 = 2 μm, for ×80,000 = 200 nm. (**d**) Cytochrome c released from FLO-1 and JH-EsoAd1 cells measured using flow cytometry 20 h post APR-246 treatment. (**e**) Viability of FLO-1 and JH-EsoAd1 cells at 96 h post treatment with APR-246 and trolox (1 mM), ferrostatin-1 (Fer-1, 20 μM) or 2-merceptoethanol (2-ME, 100 μM). One-way ANOVA with Dunnett's multiple comparison post-test (**e**). Error bars = s.e.m., $n = 3$ for all, *$P < 0.05$, **$P < 0.01$, ***$P < 0.001$. See also Supplementary Fig. 2.

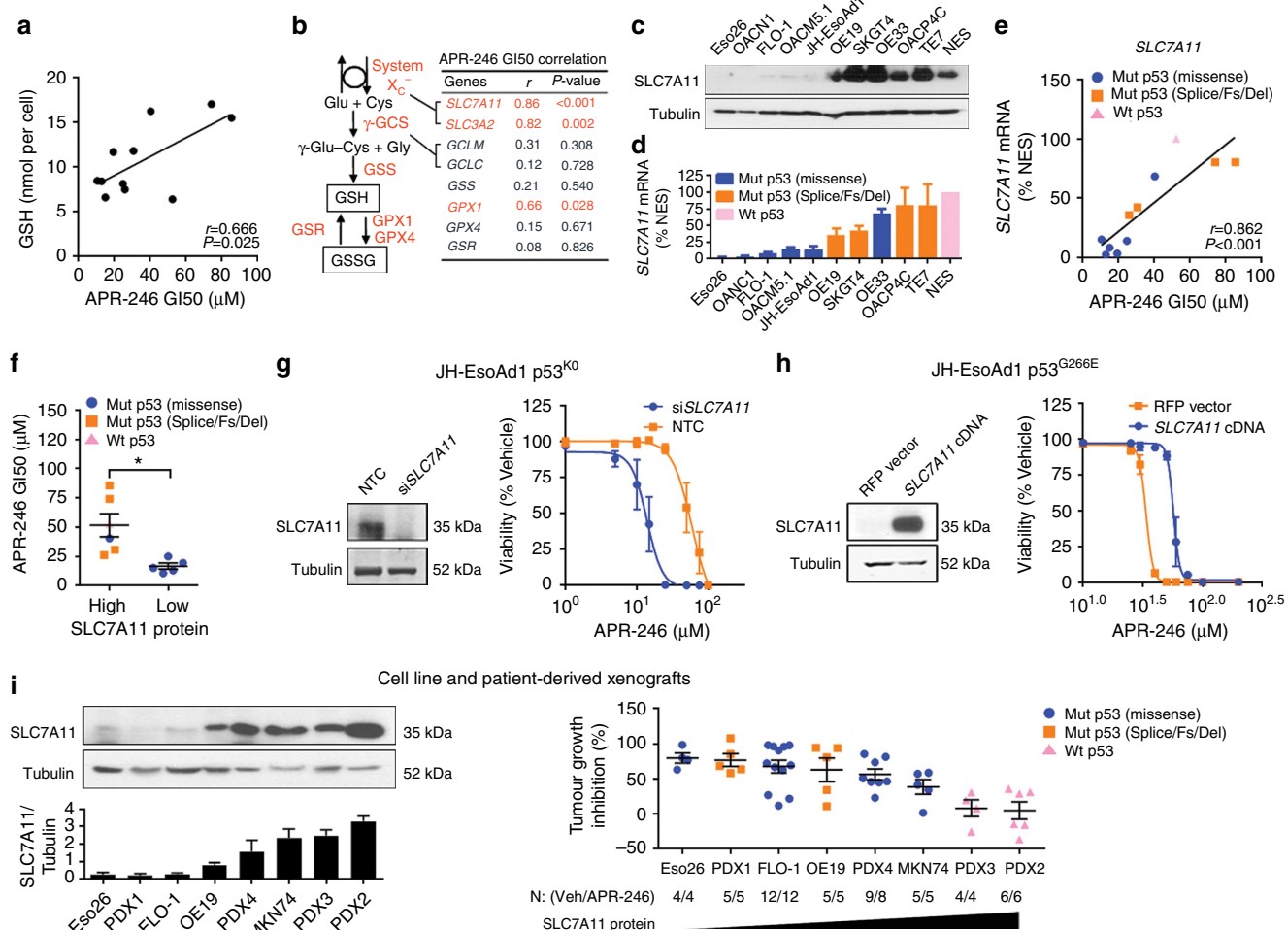

**Figure 3 | SLC7A11 expression predicts and modulates tumour sensitivity to APR-246.** (**a,b**) Correlation between endogenous glutathione (GSH) levels (**a**) and mRNA expression of GSH pathway genes (**b**) with APR-246 sensitivity (GI50) in oesophageal cell lines. See also Supplementary Table 1. (**c,d**) Endogenous protein (**c**) and mRNA (**d**) levels of *SLC7A11* in oesophageal cell lines. (**e,f**) Correlation between mRNA (**e**) and protein (**f**) levels of *SLC7A11* with APR-246 GI50 in oesophageal cell lines. (**g**) *SLC7A11* knockdown in p53[KO] JH-EsoAd1 cells. APR-246 was applied 2 days post transfection of *SLC7A11* and non-targeting control (NTC) siRNA with viability measured at 96 h post APR-246 (right). Knockdown was confirmed by western blot (left) 3 days post transfection. (**h**) *SLC7A11* and red fluorescent protein (RFP) were overexpressed in mut-p53 JH-EsoAd1 cells. This was confirmed by western blot (left). Cell viability was measured 96 h post APR-246 (right). (**i**) Correlation between SLC7A11 protein level (left, *n* = 4) and *in vivo* response to APR-246 (100 mg kg⁻¹, intraperitoneal injection, daily for 3 weeks. PDX4 was treated daily for 2 weeks) in cell line and patient-derived xenografts (PDX, right). % Tumour growth inhibition was quantified at endpoint. Pearson's correlation (**a,b,e**), unpaired *t*-test (**f**). Error bars = s.e.m., *n* = 3 for all *in vitro* studies. See also Supplementary Fig. 3.

SLC7A11 expression (Fig. 3e,f). To interrogate the functional relationship between *SLC7A11* and cellular sensitivity to APR-246, we firstly knocked down *SLC7A11* in JH-EsoAd1 p53[KO] (endogenous mut-p53 was knocked out using CRISPR/Cas9) and parental H1299 p53[Null] cells. In these cells, which are relatively resistant to APR-246 (ref. 3), we found that inhibiting *SLC7A11* expression sensitized them to APR-246 (Fig. 3g; Supplementary Fig. 3e). Contrastingly, ectopic expression of *SLC7A11* in two APR-246 sensitive mut-p53 lines, JH-EsoAd1 and FLO-1, induced resistance to APR-246 (Fig. 3h; Supplementary Fig. 3f). To corroborate these findings *in vivo*, we found that endogenous *SLC7A11* expression correlated closely with tumour response to APR-246 in eight patient-derived xenograft and cell line models (Fig. 3i). Overall, these results highlight *SLC7A11* as a novel regulator and predictive biomarker of response to APR-246.

**Mut-p53 entraps NRF2 and represses SLC7A11 expression.** Given that *SLC7A11* expression predicts and regulates tumour

sensitivity to the prototypical mut-p53 reactivator, APR-246, we next examined the relationship between mut-p53 and *SLC7A11*. We found that *SLC7A11* mRNA and protein levels inversely correlated with accumulation of mut-p53 protein across our oesophageal cell line panel (Fig. 4a,b). Consistent with this, analysis of multiple TCGA datasets correlating p53 protein (RPPA) and *SLC7A11* mRNA (RNAseq) expression demonstrated the same relationship (Supplementary Fig. 4a). In support of these findings, we observed an increase in *SLC7A11* expression following siRNA knockdown and CRISPR/Cas9 knockout of endogenous mut-p53 in FLO-1 and JH-EsoAd1 cells, respectively (Fig. 4c,d). In contrast, ectopic expression of mut-p53 in H1299 p53[Null] cells decreased *SLC7A11* mRNA and protein levels (Fig. 4e). Collectively, these results illustrate that accumulation of mut-p53 suppresses *SLC7A11* expression.

We next investigated the mechanism by which mut-p53 suppressed *SLC7A11* expression. Since mut-p53 loses its capacity to bind DNA in a sequence-specific manner, we hypothesized that direct trans-repression is an unlikely mechanism of *SLC7A11*

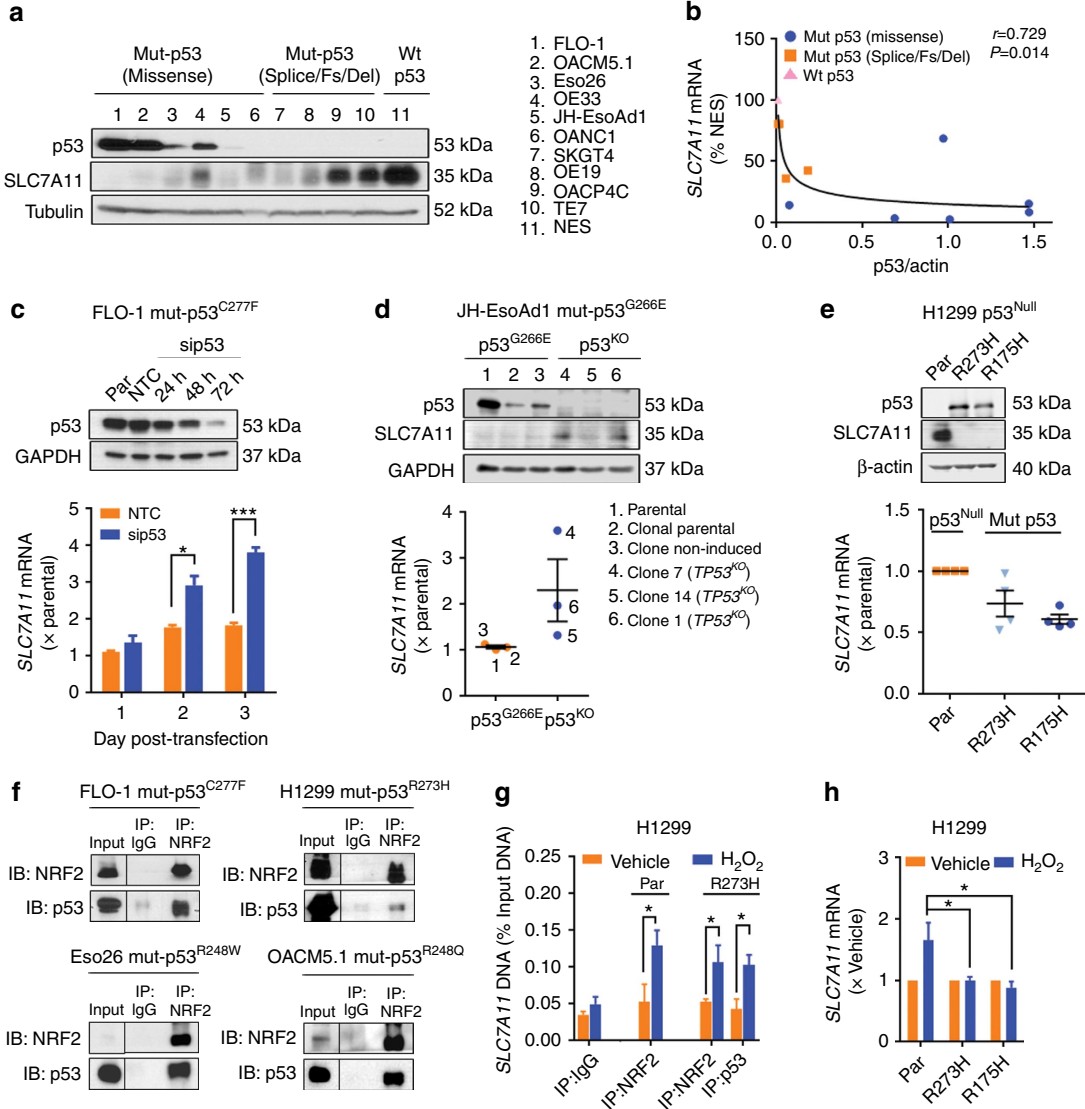

**Figure 4 | Accumulated mut-p53 entraps NRF2 and represses *SLC7A11* expression.** (**a,b**) Correlation between p53 with SLC7A11 protein (**a**) and mRNA levels (**b**) in oesophageal cell lines. (**c–e**) *SLC7A11* mRNA (bottom) and protein (top) levels post mut-p53 knockdown (**c**, non-targeting control (NTC) and p53 siRNA (sip53)) in FLO-1 cells, knockout (**d**) in JH-EsoAd1 cells and overexpression (**e**) in H1299 cells. (**f**) Immunoprecipitation (IP) and immunoblot (IB) of NRF2 with mut-p53 in FLO-1, Eso26, OACM5.1 and H1299 cells under basal growth conditions. Reverse IP and IB in Supplementary Fig. 4. (**g**) Chromatin IP of the *SLC7A11* promoter using NRF2 and mut-p53 antibodies at rest and post $H_2O_2$ stress (4 h, 50 μM) in p53^Null (Par) and p53^R273H H1299 cells. (**h**) *SLC7A11* mRNA expression following vehicle or $H_2O_2$ (50 μM) treatment in p53^Null (Par), p53^R273H and p53^R175H H1299 cells. Pearson's correlation (**b**), unpaired *t*-test (**c,g**), one-way ANOVA with Dunnett's multiple comparison post-test (**h**). Error bars = s.e.m., *n* = 3 for all except (**f**) *n* = 2, *$P < 0.05$, ***$P < 0.001$. See also Supplementary Fig. 4.

regulation by mut-p53. Seminal studies have shown that the master antioxidant transcription factor, NRF2, intimately regulates *SLC7A11* transcription[14,15]. We thus searched for evidence of potential interaction between mut-p53 and NRF2. Firstly, using genetic knockdown studies, we confirmed that NRF2 does indeed regulate *SLC7A11* transcription in our cell lines (Supplementary Fig. 4b). Furthermore, we observed that the expression of multiple NRF2 target genes responsible for maintaining cellular redox balance (for example, *NQO1*, *PRDX1*, *OGGIN1*, *HMOX1*, *KEAP1* and *SLC3A2*) increased following mut-p53 knockdown and decreased with mut-p53 overexpression (Supplementary Fig. 4c,d). In parallel, we reviewed the TCGA datasets and identified a similar inverse correlation between accumulation of mut-p53 and expression of *NQO1*, a canonical target gene of NRF2 (Supplementary Fig. 4a), suggesting that mut-p53 is able to interfere with NRF2 activity.

Extending this, given that mut-p53 knockdown, knockout and overexpression did not consistently or significantly affect the mRNA and protein levels of NRF2 (Supplementary Fig. 4e–g), we performed protein immunoprecipitation studies and demonstrated that mut-p53 can interact with NRF2 (Fig. 4f; Supplementary Fig. 4h). Moreover, this protein interaction is conserved across different tumour types (oesophageal: FLO-1, Eso26 and OACM5.1, and lung: H1299) and mut-p53 genotypes. Importantly, chromatin immunoprecipitation analysis identified that mut-p53 is recruited along with NRF2 to the *SLC7A11* promoter under oxidative stress (Fig. 4g), which in turn significantly impairs the efficiency of NRF2-mediated upregulation of *SLC7A11* (Fig. 4h).

**Mut-p53 accumulation sensitizes cancer cells to ROS stress.** We next examined the consequences of mut-p53 accumulation on

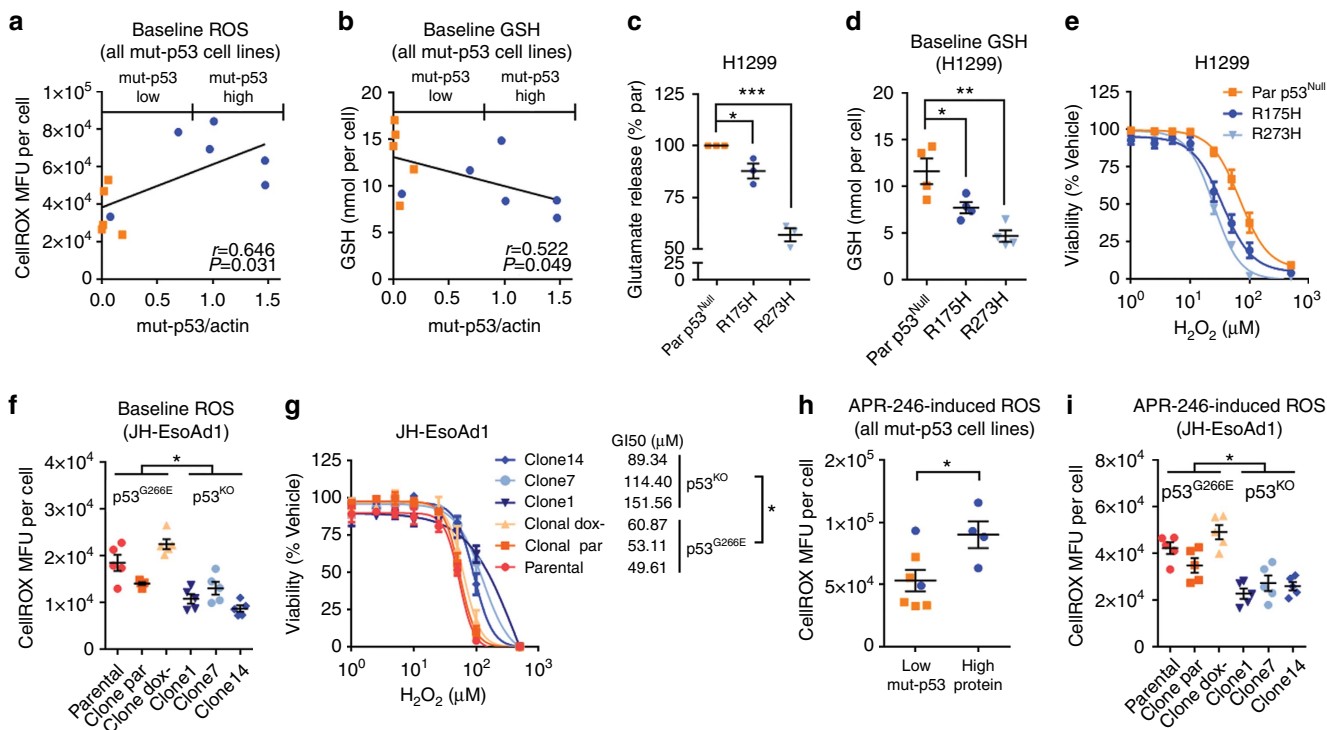

**Figure 5 | Mut-p53 accumulation sensitizes cancer cells to oxidative stress. (a,b)** Correlation of mut-p53 protein (normalized to β-actin) with ROS (**a**) and glutathione (GSH) (**b**) levels under basal growth conditions for all mut-p53 cancer cell lines used in this study. For further analysis cell lines were dichotomized into mut-p53 high and low expressers based on western blot in Fig. 4a. (**c**) System $x_C^-$ activity as assayed by glutamate release from p53[Null], p53[R273H] and p53[R175H] H1299 cells. Cells were stimulated with 600 μM L-cystine for 3 h prior to glutamate assay. (**d**) GSH levels at rest in p53[Null], p53[R273H] and p53[R175H] H1299 cells. (**e**) Cell viability at 96 h post $H_2O_2$ treatment in p53[Null], p53[R273H] and p53[R175H] H1299 cells. (**f**) ROS levels at rest in p53[G266E] and mut-p53[KO] JH-EsoAd1 cells. (**g**) Cell viability at 96 h post $H_2O_2$ treatment in p53[G266E] and mut-p53[KO] JH-EsoAd1 cells. (**h**) ROS levels 6 h post 50 μM APR-246 treatment in all mut-p53 cancer cell lines grouped by endogenous mut-p53 protein levels as determined in **a,b**. (**i**) ROS levels 6 h post 50 μM APR-246 treatment in p53[G266E] and mut-p53[KO] JH-EsoAd1 cells. Pearson's correlation (**a,b**), one-way ANOVA with Dunnett's multiple comparison post-test (**c,d**), unpaired *t*-test (**f–i**). Error bars = s.e.m., *n* = 3 for all except (**d**) *n* = 4 and (**f,i**) *n* = 5, *$P < 0.05$, **$P < 0.01$, ***$P < 0.001$. See also Supplementary Fig. 5.

cellular redox balance. Across our cell line panel, accumulation of mut-p53 protein was significantly associated with increased basal ROS levels (Fig. 5a), and decreased endogenous GSH reserves (Fig. 5b). Consistently, overexpression of mut-p53 in H1299 cells diminished system $x_C^-$ activity (Fig. 5c) and GSH levels (Fig. 5d), resulting in heightened sensitivity to ROS stress (Fig. 5e). In contrast, knockout of mut-p53 in JH-EsoAd1 cells decreased basal ROS levels (Fig. 5f) and conferred protection against $H_2O_2$ (Fig. 5g). These findings were replicated when *SLC7A11* expression was directly manipulated (Supplementary Fig. 5a–h). Moreover, we found that cells with mut-p53 accumulation had significantly higher amounts of ROS following APR-246 treatment when compared to cells with low levels or absence of mut-p53 protein (Fig. 5h,i). Overall, these results establish that accumulation of mut-p53, through entrapping NRF2 and repressing *SLC7A11* expression, sensitizes cells to oxidative stress.

**System $x_C^-$ blockade targets cancers with mut-p53 accumulation.** Since cancer cells with mut-p53 accumulation are vulnerable to oxidative stress, we next sought to identify potential antioxidative stress genes which when suppressed, would preferentially kill cancer cells with stabilized mut-p53. Using a multi-omics approach, we examined the outcomes of independently knocking down 438 antioxidative stress genes by analysing the Broad Institute's Project Achilles dataset (genome-wide shRNA knockdown in 201 cancer cell lines[16]) (see Methods). Remarkably, cell lines with missense

*TP53* mutations, which typically accumulate mut-p53 protein, are most sensitive to knockdown of *SLC7A11* than any other antioxidative stress gene (Fig. 6a; Supplementary Data 1). Importantly, this finding was consistent across different tumour types (Fig. 6b). To further validate this, we knocked down *SLC7A11* using siRNA in our oesophageal cell line panel (Fig. 6c), and demonstrated significant impairment of viability and clonogenicity of cells with missense *TP53* mutations, which are predominantly p53 high and *SLC7A11* low expressers (Fig. 6d,e; Supplementary Fig. 6a). In addition, we showed that erastin and sulfasalazine (SAS), proven inhibitors of system $x_C^-$ (ref. 9) (Supplementary Fig. 6b,c), phenocopied our genetic ablation studies (Fig. 6f; Supplementary Fig. 6d,f). Moreover, mut-p53 overexpression enhanced the sensitivity of p53[Null] H1299 cells to *SLC7A11* knockdown and to system $x_C^-$ inhibitors (Fig. 6g,h; Supplementary Fig. 6g), while knockout of mut-p53 in JH-EsoAd1 cells had the opposite effect (Supplementary Fig. 6h,i). Altogether, these results illustrate that system $x_C^-$ antagonists can preferentially kill cancer cells that accumulate mut-p53 protein.

**SLC7A11 blockade and APR-246 synergize to kill mut-p53 tumours.** Since APR-246 depletes GSH and system $x_C^-$ blockade leads to cystine starvation, which in turn impairs GSH synthesis, we hypothesized that their therapeutic combination would synergistically target mut-p53 cancer cells. To test this approach, we applied APR-246 to our oesophageal cell lines following *SLC7A11* knockdown. We found that inhibiting *SLC7A11*

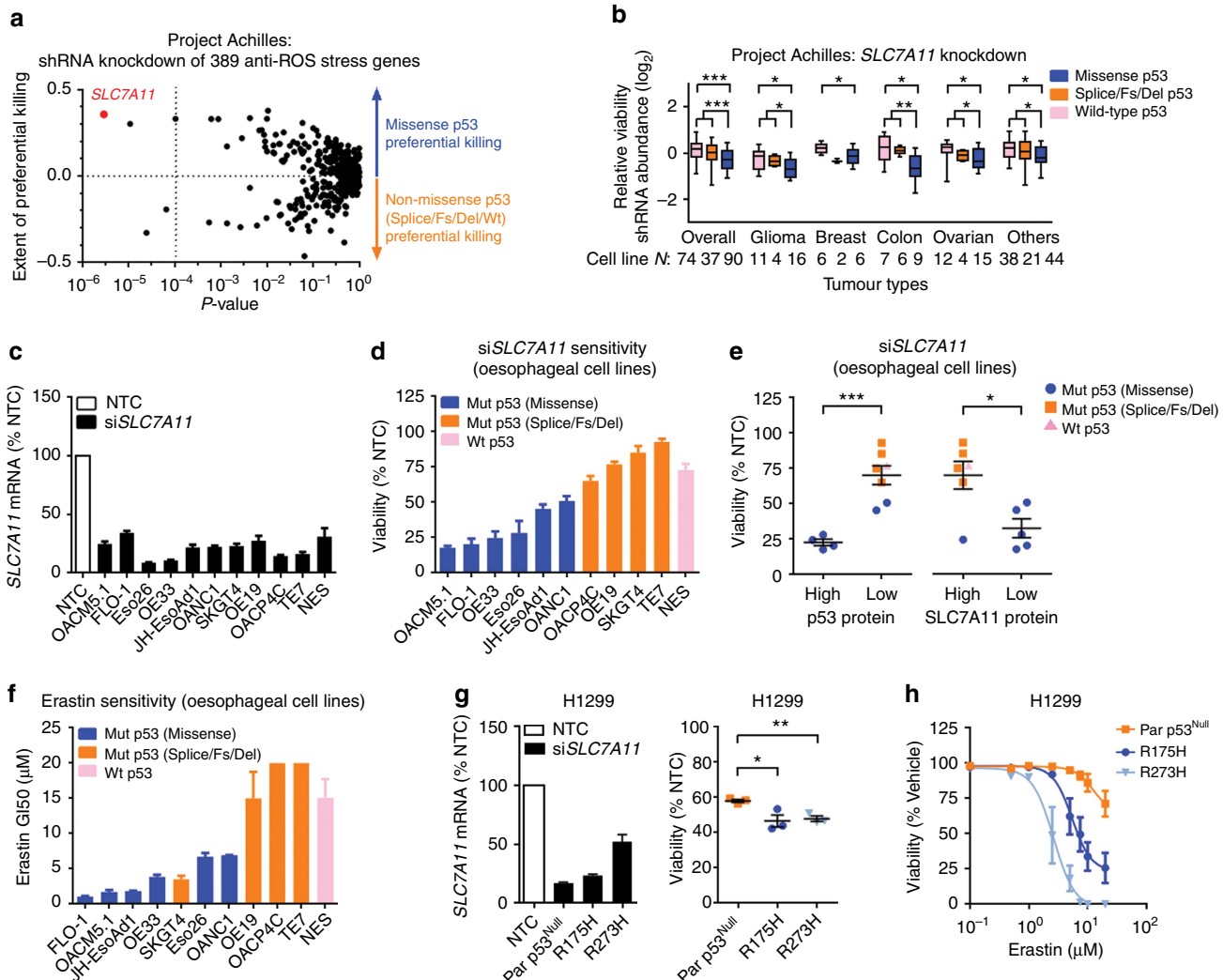

**Figure 6 | System x$_C^-$ inhibition selectively targets cancer cells with mut-p53 accumulation.** (**a**) Plot of 389 anti-ROS stress genes identified in an unbiased multi-omics analysis of the Broad Institute's Project Achilles v2.4 dataset (See Methods and Supplementary Fig. 9 for analysis workflow), correlating statistical significance with cancer cell killing following shRNA-mediated gene knockdown. Vertical axis quantifies the extent of preferential killing of missense mut-p53 cells (which typically accumulate mut-p53 protein) versus non-missense p53 cells (including wt-p53, splice, frameshift (Fs) and deletion (Del) variants). Vertical dotted line: Bonferroni corrected $P$-value $= 1.1 \times 10^{-4}$ comparing the effect of gene knockdown on missense mut-p53 versus non-missense p53 cancer cells. See also Supplementary Data 1. (**b**) Cell viability following shRNA-mediated knockdown of *SLC7A11* in cell lines from Project Achilles v2.4. Each box/whiskers plot = median/5–95 percentiles. (**c**) *SLC7A11* mRNA levels 48 h post transfection with non-targeting control (NTC) siRNA or si*SLC7A11* in oesophageal cell lines. (**d,e**) Viability at 96 h post *SLC7A11* knockdown (**d**) and grouped by each cell line's endogenous p53 (**e**, left) or SLC7A11 (**e**, right) protein expression. (**f**) Erastin sensitivity (GI50) in oesophageal cell lines. (**g**) *SLC7A11* knockdown in parental (Par) p53$^{Null}$ and mut-p53 overexpressing H1299 cells. *SLC7A11* mRNA (left) and viability (right) was measured at 48 and 96 h post siRNA transfection, respectively. (**h**) Viability at 96 h post erastin treatment in p53$^{Null}$ and mut-p53 overexpressing H1299 cells. For all studies $n = 3$ except (**h**) $n = 5$, and (**a,b**). Unpaired $t$-test (**b,e**), one-way ANOVA with Dunnett's multiple comparison post-test (**g**). Error bars = s.e.m., *$P < 0.05$, **$P < 0.01$, ***$P < 0.01$. See also Supplementary Fig. 6.

expression significantly enhanced the efficacy of APR-246, particularly against cancer cells with mut-p53 accumulation (Fig. 7a), resulting in synergistic induction of ROS and apoptosis (Fig. 7b,c; Supplementary Fig. 7a). Furthermore, both SAS and erastin strongly synergized with APR-246 to inhibit mut-p53 cancer cells. This effect was most pronounced in cells with missense *TP53* mutations expressing high mut-p53 protein (Fig. 7d,e; Supplementary Fig. 7b). To validate that this synergistic interaction is indeed dependent on mut-p53 and *SLC7A11* expression, we next demonstrated that these drug combinations were more effective in H1299 cells overexpressing mut-p53 compared to the p53$^{Null}$ parental line (Fig. 7f; Supplementary Fig. 7c). In contrast,

overexpression of *SLC7A11* in mut-p53 FLO-1 cells abolished the synergistic activity between system x$_C^-$ inhibitors and APR-246 (Fig. 7g; Supplementary Fig. 7d). Mechanistically, we showed that system x$_C^-$ blockade in conjunction with APR-246, synergistically depleted intracellular GSH, resulting in mitochondrial ROS accumulation, lipid peroxidation and ultimately apoptotic cell death (Fig. 7h–k; Supplementary Fig. 7e–h).

To extend these findings *in vivo*, we generated inducible sh*SLC7A11* constructs in OE33 and FLO-1 cells using a dual fluorescence reporter system (Fig. 8a; Supplementary Fig. 8a). These hairpins efficiently knocked down *SLC7A11* and had significant inhibitory activity alone or in combination with

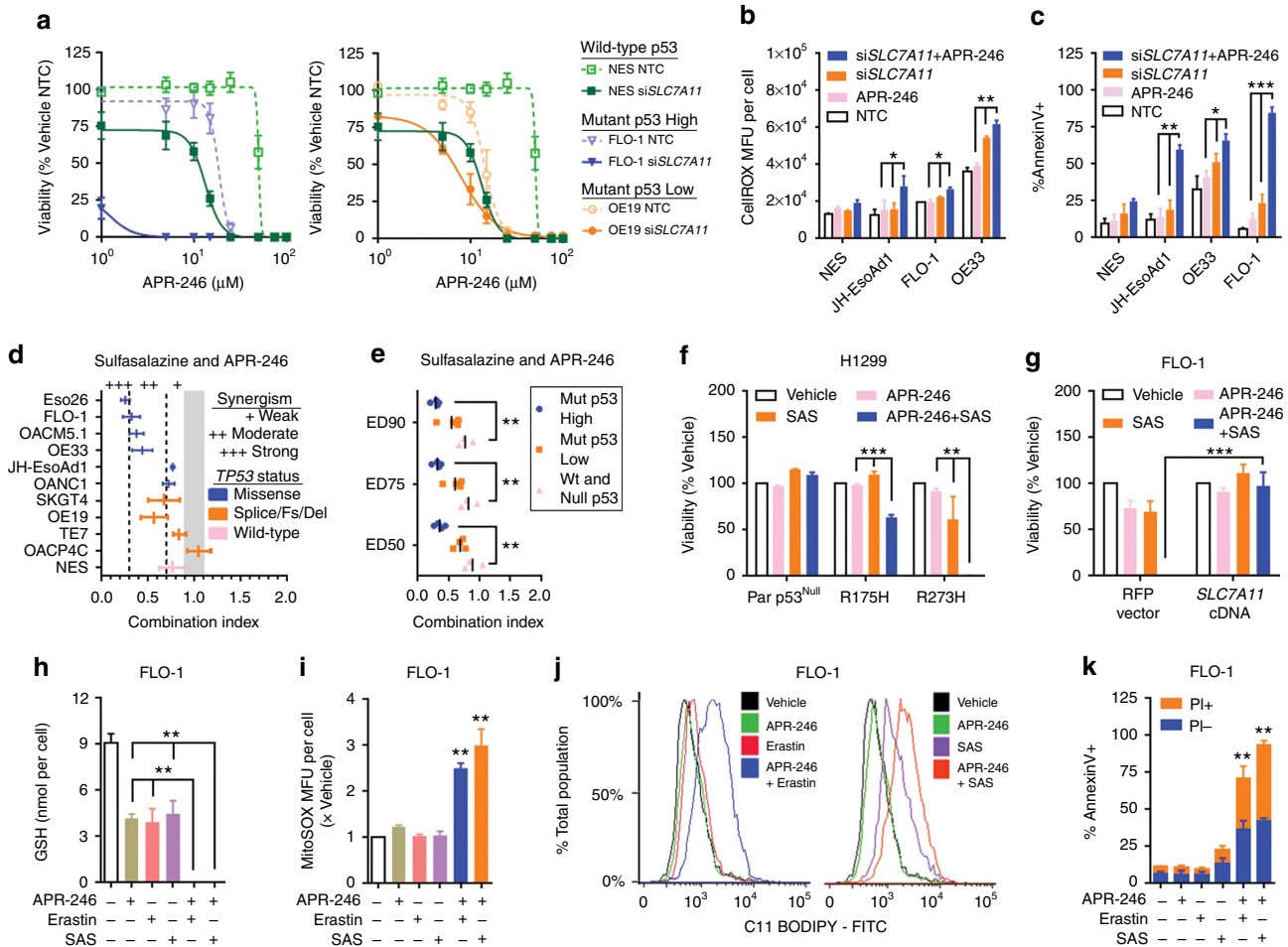

**Figure 7 | System $x_C^-$ antagonists synergize with APR-246 to inhibit mut-p53 cancer cells.** (**a**) Viability at 96 h post APR-246 treatment in non-targeting control (NTC) or si*SLC7A11* transfected mut-p53 high (left) and low (right) expressing cell lines. Cells were transfected with siRNA 48 h prior to APR-246. (**b,c**) CellROX (**b**) and Annexin-V (**c**) analysis in cells treated with 10 µM APR-246 48 h post NTC or si*SLC7A11* transfection. CellROX and Annexin-V were assayed at 24 and 48 h post APR-246 respectively. Mean fluorescence unit (MFU). (**d**) Combination index (CI) plot of cell lines treated with APR-246 and sulfasalazine (SAS) for 96 h. For CI analysis, cells were treated with a range of doses for each agent alone and in combination. Synergistic interaction was quantified using CalcuSyn v2, where a CI < 0.9: synergism, CI > 1.1: antagonism and $0.9 \le CI \le 1.1$ (Grey area): additive effect. CI plot shows the extent of drug interaction to achieve 50% cell death. (**e**) Extent of synergistic interaction between SAS and APR-246 to achieve 50% (ED50), 75% (ED75) and 90% (ED90) cell death in oesophageal cell lines with different levels of p53 protein. Each point = mean CI per cell line ($n = 3$). Bars = mean of each group. (**f**) Viability at 96 h post treatment with 10 µM APR-246 and/or 400 µM SAS in parental (Par) p53$^{Null}$ and mut-p53 overexpressing H1299 cells. (**g**) Viability at 96 h post treatment with 7.5 µM APR-246 and/or 200 µM SAS in FLO-1 cells with either *SLC7A11* or red fluorescence protein (RFP) overexpression. (**h–k**) GSH (**h**), MitoSOX (**i**), C11-BODIPY (**j**) and Annexin-V/PI (**k**) analysis of FLO-1 cells treated with 7.5 µM APR-246 and/or 0.5 µM erastin or 200 µM SAS. GSH, MitoSOX, C11-BODIPY and Annexin-V/PI were assayed at 15, 24, 24 and 48 h post treatment, respectively. For **b,c,f–k** the dose of APR-246, erastin or SAS was deliberately chosen to have low cytotoxicity on its own to highlight the combinatory effect. One-way ANOVA with Dunnett's multiple comparison post-test (**b,c,e,f,h,i,k**), unpaired *t*-test (**g**). Error bars = s.e.m., $n = 3$ for all, *$P < 0.05$, **$P < 0.01$, ***$P < 0.01$. See also Supplementary Fig. 7.

APR-246 *in vitro* (Supplementary Fig. 8b–j). In FLO-1 xenografts, we found that *SLC7A11* knockdown significantly enhanced the anti-tumour activity of APR-246 (Fig. 8b,c; Supplementary Fig. 8k), leading to synergistic reduction of intratumoral GSH levels and improved animal survival (Fig. 8d,e). Consistent with this, the combination of SAS with APR-246 at tolerated doses (Supplementary Fig. 8l,m) also conferred greater anti-tumour activity than single agents alone (Fig. 8f). Importantly, these findings were reproduced in a PDX model of mut-p53 high-expressing oesophageal cancer (Fig. 8g–i; Supplementary Fig. 8n–o). Analysis of these tumours revealed markedly reduced proliferation and increased apoptosis (Fig. 8j). Collectively, these studies demonstrate the therapeutic potential of combining system $x_C^-$ inhibitors with APR-246 to synergistically target cancer cells with mut-p53 accumulation.

## Discussion

Here, we propose a new paradigm for targeting mut-p53 cancers based on specific perturbations of the system $x_C^-$ and GSH axis. Our results highlight four key aspects.

First, we demonstrated that APR-246, the prototypical mut-p53 reactivator, induces lipid peroxidative cell death through GSH depletion. Although previous studies reported that APR-246 can deplete GSH[4,17], the mechanism and consequences of this has not been fully elucidated until now. Here, we demonstrated that MQ, the active component of APR-246 (ref. 18), reacts with GSH in cancer cells to form a thio-ether that is non-reducible by GSH reductase, a crucial step in the recycling of functional GSH. Ultimately, this leads to ROS accumulation, particularly in the mitochondria, resulting in lipid peroxidation, mitochondrial rupture and the release of cytochrome $c$, triggering apoptosis.

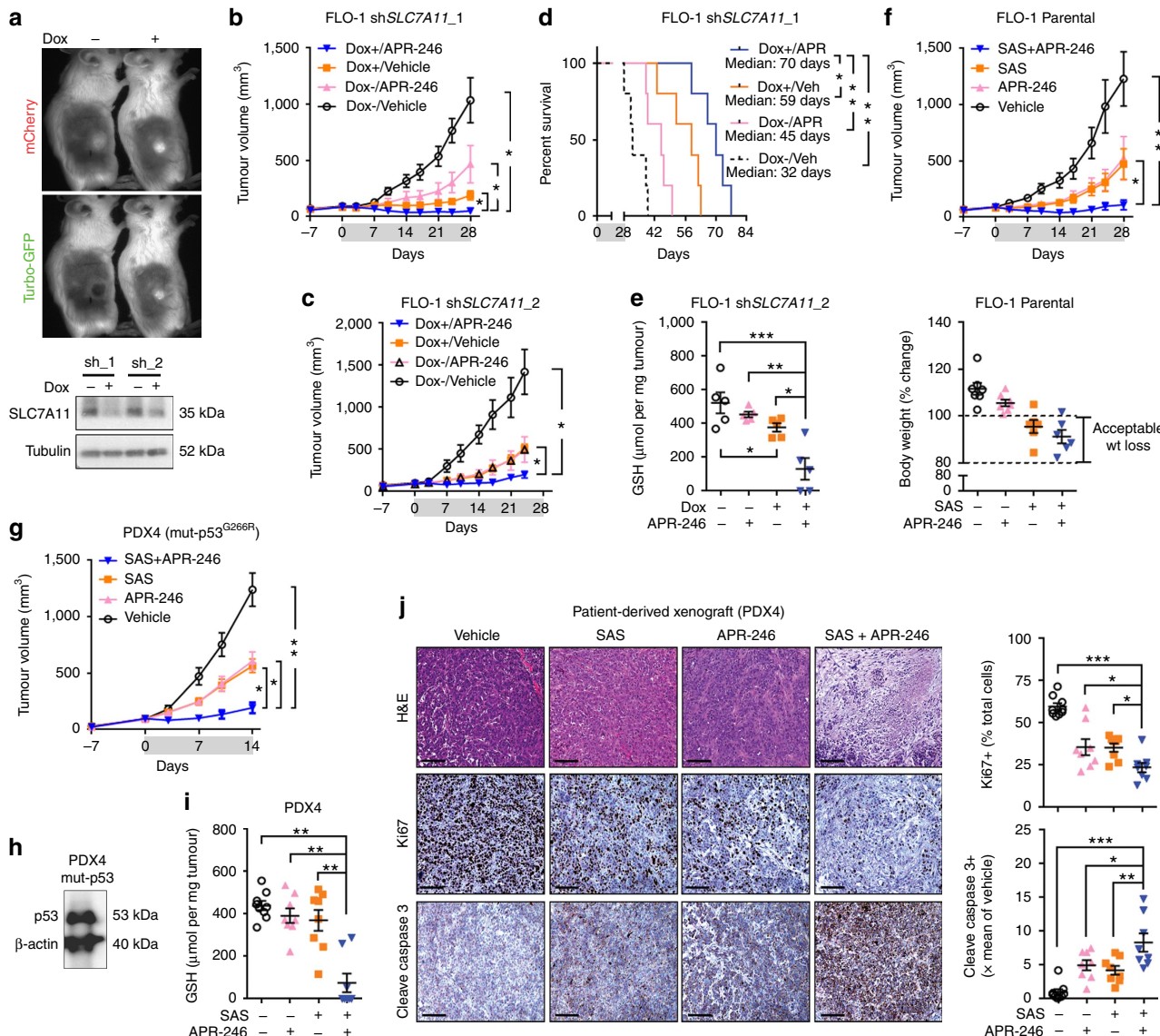

**Figure 8 | System x$_C^-$ antagonists synergize with APR-246 *in vivo*.** (**a**) Example fluorescence imaging of mice bearing FLO-1 xenografts transduced with a doxycycline (dox)-inducible dual fluorescent vector with sh*SLC7A11* cloned in (top). Western blot of FLO-1 xenografts 72 h post dox (Food: 600 mg kg$^{-1}$, water: 2 mg ml$^{-1}$) induction to assess the extent of knockdown using two hairpins (sh) against *SLC7A11* (Bottom). (**b,c**) Growth curves of FLO-1 tumours transduced with sh*SLC7A11*_1 (**b**) and sh*SLC7A11*_2 (**c**) hairpins in mice treated with APR-246 (100 mg kg$^{-1}$, daily) and/or dox for 28 days. $n = 5$ per group. (**d**) Kaplan–Meier plot of time to reach 1,500 mm$^3$ tumour volume in mice bearing FLO-1 tumours transduced with sh*SLC7A11*_1 and treated according to conditions detailed in **b**. (**e**) Intratumoral GSH levels in FLO-1 xenografts transduced with sh*SLC7A11*_2 at 28 days post treatment. (**f**) Growth curves of FLO-1 xenografts in mice treated with either vehicle (0.9% saline, daily, $n = 7$), APR-246 (100 mg kg$^{-1}$, daily, $n = 7$), sulfasalazine (SAS, 6 mg, twice daily, $n = 6$) or APR-246 and SAS ($n = 6$) for 28 days (top). Body weight at 28 days post treatment as a percentage change from baseline (bottom). Ethically acceptable weight loss is defined by the Peter MacCallum Cancer Centre Animal Experimentation Ethics Committee as <20% compared to pre-treatment body weight (within the dotted lines). (**g**) Growth curves of Patient-derived xenograft 4 (PDX4) in mice treated with the same drug regimen as (**f**) for 14 days. Vehicle: $n = 9$, all other groups: $n = 8$. (**h**) Representative western blot of PDX4 demonstrating the accumulation of mut-p53 protein. (**i**) Intratumoral GSH levels in PDX4 at 14 days post treatment with drug regimens detailed in **g**. (**j**) Representative H&E, Ki67 and Cleave caspase 3 staining of PDX4 tumours (left) and their respective quantification (right). Scale bars = 100 µm. Grey shading: treatment period. One-way ANOVA with Dunnett's multiple comparison post-test (**b–g,i,j**). Error bars = s.e.m., $*P < 0.05$, $**P < 0.01$, $***P < 0.01$. See also Supplementary Fig. 8.

Given that GSH is the predominant ROS-scavenger in cells, and the source of reducing equivalents for many redox modulating enzymes such as peroxidases, peroxiredoxins and thiol reductases[7], the loss of GSH significantly impairs cellular response to oxidative stress. Furthermore, recent studies have demonstrated that GSH depletion can directly activate lipoxygenases and suppress GPX4 activity to trigger lipid peroxidation and apoptosis[19,20]. In summary, GSH depletion

and lipid peroxidation are integral to the anti-tumour activity of APR-246.

Second, we identified that expression of *SLC7A11*, a key component of system x$_C^-$ that imports cystine for the formation of GSH, is a novel and robust predictive biomarker of tumour response to APR-246. Specifically, cells with low *SLC7A11* levels proved the most sensitive to APR-246. In this way, our study unifies two key aspects of APR-246 activity: redox modulation

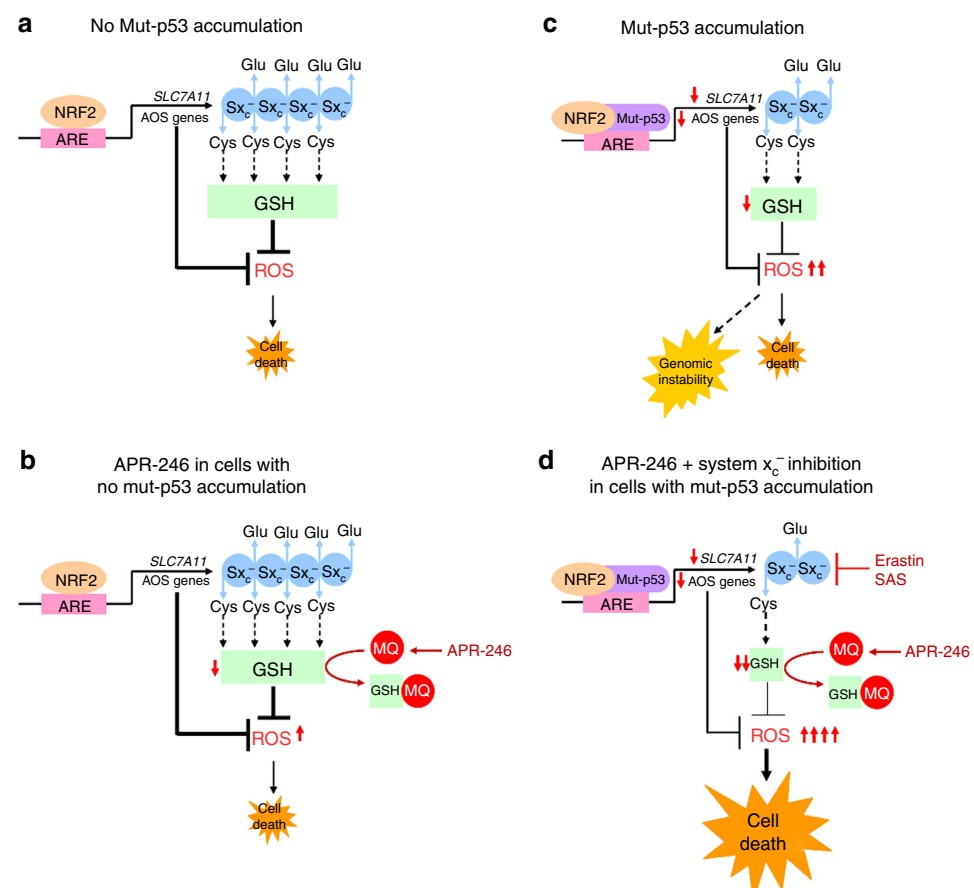

**Figure 9 | Model representation of mut-p53 entrapment of NRF2 and its implications for cellular redox balance and therapeutic intervention. (a)** In the absence of mut-p53 accumulation, NRF2 is able to transcriptionally regulate cellular redox balance by binding to antioxidant responsive elements (ARE) on antioxidative stress (AOS) genes. A crucial component of NRF2-mediated redox regulation is the transactivation of *SLC7A11*, a key component of the glutamate (Glu)/cystine (Cys) antiporter, system $x_c^-$ ($Sx_c^-$), resulting in maintenance of intracellular glutathione (GSH) reserves. **(b)** Cells with no mut-p53 accumulation have sufficient GSH reserves, and can mount a normal NRF2-mediated defence against oxidative stress, such as that induced by APR-246 (active compound: MQ). Therefore, these cells are relatively resistant to APR-246. **(c)** In cancer cells with mut-p53 accumulation, mut-p53 entraps NRF2 and impairs its canonical transcriptional activity, resulting in suppressed expression of *SLC7A11* and other AOS genes. This reduces GSH reserves and increases resting levels of ROS. Although these cells are tolerant of this, they are susceptible to further oxidative stress and genomic instability. **(d)** Cancer cells with mut-p53 accumulation are therefore highly sensitive to system $x_c^-$ inhibitors (for example sulfasalazine (SAS) and erastin) and APR-246. In combination, these agents synergistically deplete mut-p53 cancer cells of GSH, resulting in significant oxidative stress and massive cell death.

and mut-p53-dependent killing of cancer cells[21]. Our finding that *SLC7A11* knockdown in p53[Null] models sensitized cells to APR-246, while *SLC7A11* overexpression in mut-p53 models generated drug resistance, indicates that the therapeutic activity of APR-246 is dependent not only upon mut-p53 status, but also on the basal level of *SLC7A11* expression and intracellular GSH.

Our studies elucidated a novel and fundamental reciprocal relationship between *SLC7A11* expression and mut-p53 accumulation (Fig. 9). Critically, we demonstrated that suppression of *SLC7A11* transcription results from mut-p53 binding and interfering with the activity of the master antioxidant transcription factor NRF2, a mechanism which is separate from the transcriptional regulation of *SLC7A11* by wt-p53 described by Jiang *et al.*[22]. Through mut-p53 deregulation of *SLC7A11* and consequent depletion of intracellular GSH, leading to ROS accumulation, cells are sensitized to APR-246. In contrast, cells that are p53 null or wild-type have relatively higher levels of *SLC7A11* expression, and are therefore resistant to APR-246 treatment. In this way, our study resolves the apparent conflict within the literature reporting variability in p53-dependent activity and the extent of functional p53 reactivation by APR-246 (refs 17,23,24).

An interesting additional perspective related to these findings is that overexpression of c-Myc, which is a known up-regulator of *SLC7A11* (ref. 25), has recently been found to induce APR-246 resistance[26]. Similarly, we expect that other factors that modulate *SLC7A11* levels, such as NRF2 (ref. 27), ATF4 (ref. 14) and CD44v (ref. 28), may also influence drug response. There is a worldwide consensus that reliable predictive biomarkers are critical to the success of personalized cancer medicine[29]. With new clinical trials of APR-246 commencing in multiple tumour streams[2], our discovery of *SLC7A11* expression as a robust companion biomarker, in addition to mut-p53 status, for APR-246, may facilitate selection and enrichment of patients with mut-p53 tumours who are most likely to respond to APR-246 therapy.

Third, we discovered that mut-p53 accumulation sensitizes cancer cells to oxidative stress, secondary to impaired NRF2 activity, and that these cancer cells can be selectively targeted using system $x_c^-$ inhibitors. There is increasing evidence that aberrant accumulation of mut-p53 leads to gain-of-function activities in cancer cells[30]. In line with this, a recent study found that missense p53 mutants cooperate with NRF2 to inhibit multiple anti-tumour pathways in cancer cells by upregulating the proteasome machinery[31]. Here, we demonstrate that this

gain-of-function programme has a significant weakness. By binding to NRF2 and suppressing its canonical transcriptional ability to regulate cellular redox balance, mut-p53 directly decreases GSH reserves and increases basal ROS levels, thereby predisposing cancer cells to further oxidative stress (Fig. 9a,c). While a moderate increase in ROS is tolerated and may even be advantageous for survival[6], excessive ROS accumulation leads to cell death[7]. Since normal cells can mount an optimal NRF2-mediated response to oxidative stress (Fig. 9b), we propose that accumulation of mut-p53 in cancer cells, through its repressive effects on NRF2 activity and SLC7A11 expression, creates an 'Achilles heel' that can be exploited by further inhibition of system $x_C^-$ (Fig. 9d). This approach differs from treating oncogenic addiction[32], whereby cancer cells are sensitive to the inactivation of an overexpressed oncogenic driver. Indeed, we clearly demonstrate that cancer cells with high SLC7A11 levels were the least responsive to system $x_C^-$ inhibition. Importantly, the Broad Institute's Project Achilles data indicate that knockdown of SLC7A11, but not any other antioxidative stress genes, results in killing of cancer cells with stabilized mut-p53. This suggests that beyond supplying cystine for GSH synthesis, system $x_C^-$ may participate in other processes that mut-p53 is also reliant upon. Therefore, elucidating these processes and how they can be exploited will aid in developing further strategies to target mut-p53 cancer cells.

Fourth, having identified that mut-p53 cancer cells are highly sensitive to system $x_C^-$ inhibition, we next rationalized novel drug combinations to enhance mut-p53 specific killing of cancer cells. Here, we demonstrated that combining system $x_C^-$ blockade with APR-246 at relatively low doses, selectively and synergistically depleted mut-p53 cancer cells of GSH, leading to significant lipid peroxidation and induction of massive apoptosis (Fig. 9d). On the basis of these findings, we propose that other system $x_C^-$ inhibitors such as erastin analogs[33] and sorafenib[34], should also effectively synergize with APR-246. Indeed, in a recently completed Phase I trial of APR-246 in patients with haematological malignancies, it was noted that the addition of APR-246 to a patient receiving concurrent sorafenib resulted in complete remission of their acute myeloid leukaemia[35]. This provides proof-of-concept that combining system $x_C^-$ inhibitors with APR-246 is clinically translatable and may potentially have significant therapeutic efficacy against mut-p53 malignancies.

In summary (Fig. 9), our study demonstrates that accumulation of mut-p53 entraps NRF2 and represses SLC7A11 expression. This renders mut-p53 cancer cells susceptible to oxidative stress. As single agents, APR-246 and system $x_C^-$ inhibitors exploit this vulnerability to selectively target mut-p53 cancer cells in a synthetic lethal-like fashion. In combination, these agents synergize to further deplete GSH, leading to overwhelming ROS accumulation and cell death. Moreover, we demonstrate that SLC7A11 expression is a reliable predictive biomarker for tumour response to these therapies. This, and the fact that APR-246 and system $x_C^-$ inhibitors have been safely trialled in humans[36,37], should facilitate rapid translation of our findings to improve treatment outcomes for patients with mut-p53 cancers.

## Methods

**Compounds and reagents.** APR-246 and GS-MQ were provided by Aprea Therapeutics (Solna, Sweden). Cisplatin, 5-fluorouracil, epirubicin, irinotecan and paclitaxel were from Hospira. GSH-MEE was from Cayman Chemicals. 2-mer-ceptoethanol and $H_2O_2$ were from Merck. Erastin was from SelleckChem. GSH, NAC, trolox, ferrostatin-1, Q-VD, DFO and SAS were from Sigma-Aldrich.

**Parental cell lines.** HEK-293T, H1299, OE33 and OE19 cells were from ATCC. FLO-1, OACM5.1, Eso26, SKGT4, OACP4C and TE7 cells were provided by Rebecca Fitzgerald (University of Cambridge, UK). JH-EsoAd1 cells were a gift from James Eshleman (John Hopkins University, MD, USA). MKN74 cells were

obtained from Alex Boussioutas (Peter MacCallum Cancer Centre, Australia). Immortalized human oesophageal epithelial cells (NES) were provided by Rhonda Souza (University of Texas Southwestern Medical Centre, TX, USA). OANC1 cells were derived in our laboratory[38]. TE7 and OACP4C were classified as p53$^{Null}$ controls (Supplementary Table 1). All cell lines were authenticated by short tandem repeat analysis using the PowerPlex 16 genotyping system (Promega) and confirmed mycoplasma free by PCR (Cerberus Sciences, Australia).

**Cell cultures.** All cells were maintained at 37 °C with 5% $CO_2$. Unless otherwise specified, all culture media contained 10% fetal bovine serum supplemented with 50 U ml$^{-1}$ penicillin and 50 mg ml$^{-1}$ streptomycin (Life Technologies). NES cells were maintained in modified MCDB-153 medium[3]. HEK-293T, FLO-1 and OANC1 cells were grown in DMEM containing 2.5 mM L-glutamine and 4.5 g l$^{-1}$ D-glucose (Life Technologies). All remaining cell lines were cultured in RPMI 1640 medium containing 2.5 mM L-glutamine (Life Technologies). Unless otherwise specified, the following seeding densities per well/dish were used: 96-well plates: $5 \times 10^3$ cells (FLO-1, SKGT4 and OE33), $1.5 \times 10^4$ cells (OE19) and $1 \times 10^4$ cells (all others). Twenty-four-well plates: $2.5 \times 10^4$ cells (FLO-1, SKGT4 and OE33) and $5 \times 10^4$ cells (all others). Six-well plates: $1.3 \times 10^5$ cells (FLO-1, SKGT4 and OE33) and $2.6 \times 10^5$ cells (all others). Overall, 6 cm dishes: $2 \times 10^5$ cells (FLO-1, SKGT4 and OE33) and $4 \times 10^5$ cells (all others). Overall, 10 cm dishes: $4 \times 10^5$ cells (FLO-1, SKGT4 and OE33) and $8 \times 10^5$ cells (all others).

**TP53 mutational analysis by Sanger sequencing.** Genomic DNA was extracted using the QIAamp DNA Blood Mini kit (Qiagen). Cycle sequencing was performed on a 3130 Genetic Analyzer using the BigDye Terminator v3.1 kit (Life Technol-ogies). PCR conditions and primer sequences are detailed in Supplementary Table 2.

**Western blot analysis.** Homogenized tissues or cells were lysed at 4 °C in RIPA buffer (1 mM EDTA, 1% NP-40, 0.5% sodium deoxychlorate, 0.1% SDS, 50 mM sodium fluoride, 1 mM sodium pyrophosphate in PBS) mixed with protease and phosphatase inhibitors (Roche). Equal amounts of protein were boiled, resolved by SDS-PAGE and transferred to polyvinylidene difluoride membranes. Membranes were incubated in blocking buffer (5% skim milk, 0.1% Tween20 in Tris-buffered saline) for 1 h and probed overnight with primary antibody at 4 °C. Blots were rinsed thrice (0.1% Tween20 in Tris-buffered saline, 5 min each), followed by incubation with peroxidase-conjugated secondary antibody (Dako) for 2 h at room temperature. Proteins were detected using Western Lightening Enhanced Chemi-luminescence (PerkinElmer) or ECL Plus Western blotting substrate kit (Ther-moFisher Scientific). Antibodies are detailed in Supplementary Table 3. Protein densitometric analysis was undertaken using ImageJ software (http://ima-gej.nih.gov/ij/). Extended western blots are available in Supplementary Fig. 10.

**Gene expression with quantitative RT-PCR.** Following RNA extraction (RNeasy kit, Qiagen), reverse transcription (Transcriptor First Strand cDNA Synthesis Kit, Roche) and SYBR-green RT-PCR (Lightcycler 480, Roche), gene expression was normalized to GAPDH and analysed using the $\Delta\Delta C_t$ method. Primer sequences are detailed in Supplementary Table 4.

**Mutant-p53 overexpression.** p53$^{Null}$ H1299 cells were transduced to stably express either the R273H or R175H p53 mutants as per established protocols[39]. Single cell clones were generated following G418 selection to ensure uniform mutant-p53 expression by western blotting.

**TP53 knockout using CRISPR/Cas9 technology.** TP53 knockout JH-EsoAd1 cell line was generated as previously described[3]. Briefly, a dual lentiviral vector expression system consisting of a constitutive Cas9 endonuclease (FUCas9) and an inducible sgRNA (FGT1-UTG) vector, linked to a mCherry and GFP reporter respectively, were utilized to perform CRISPR/Cas9-mediated TP53 knockout. Two sgRNA sequences were designed to target exon 4 (GGCAGCTACGGTTTC CGTCT) and 5 (GAGCGCTGCTCAGATAGCGA) of TP53 using MIT CRISPR software (http://crispr.mit.edu) and subsequently cloned into BsmBI restriction sites on FGT1-UTG. Lentiviral particles were produced from HEK-293T cells using Lenti-X (Clontech) packaging mix according to manufacturer's instructions. JH-EsoAd1 cells were transduced with FUCas9 and sorted for mCherry-positive cells followed by transduction with FGT1-UTG and sorted for GFP-positive cells. sgRNA induction with doxycycline (2 μg ml$^{-1}$) was undertaken for 72 h before single cell sorting to generate clonal cell lines for each sgRNA. p53 status was verified by western blot and TP53 knockout confirmed by MiSeq sequencing.

**SLC7A11 knockdown using shRNA.** Short hairpin mediated knockdown of SLC7A11 in FLO-1 and OE33 cells were performed by cloning SLC7A11-specific shRNA into a lentiviral tetracycline-inducible expression vector containing the optimized miR-E backbone (LT3GECIR)[40]. This system incorporates the mCherry and turbo-GFP reporters for transduction and induction respectively. FLO-1 and OE33 cells were transduced with LT3GECIR, sorted for mCherry-positive cells and

induced with doxycycline ($2 \mu g\,ml^{-1}$) for 72 h. The 2 (of 8) most efficient shRNA constructs (TGGAGTTATGCAGCTAATT and GAGGTCATTACACATATAT), as determined by FACS, RT-PCR and western blotting, were chosen for further experiments. A hairpin targeting the *renilla* gene was also constructed as above for use as a control.

**SLC7A11 overexpression.** *SLC7A11* was ectopically expressed in FLO-1, JH-EsoAd1 and H1299 cells using the GE Dharmacon Precision LentiORF pLOC lentiviral vector. In this system, the open reading frame *SLC7A11* has been cloned downstream of the CMV promoter and contains turbo-GFP as a reporter gene. The turbo-RFP gene in place of *SLC7A11* was used as a control. Following transduction, GFP-positive FLO-1, JH-EsoAd1 and H1299 cells were sorted. The expression of *SLC7A11* was measured using RT-PCR and western blotting.

**Genetic knockdown using siRNA.** Cells were reverse transfected with 40 nM p53, NRF2, *SLC7A11* or non-targeting control siRNA pools (siGenome Smartpool, Dharmacon) using Lipofectamine RNAiMax solution (Life Technologies) according to the manufacturer's guidelines. Knockdown efficiency was assessed by RT-PCR and western blotting. siRNA sequences are detailed in Supplementary Table 5.

**Cell viability assay.** Cell viability studies were conducted in 96-well format and assayed using AlamarBlue (Life Technologies) reagent as per the manufacturer's instructions. Fluorescence was measured using a FLUOstar OPTIMA microplate reader (BMG Labtech).

**Clonogenic survival assay.** FLO-1, JH-EsoAd1, OACP4C and TE7 cells were seeded at $1 \times 10^3$ cells per well in 6-well plates. OACP4C were cultured for 10 days, whilst the others were grown for 7 days, followed by fixing with 6% glutaraldehyde and staining with 0.5% crystal violet. Discrete colonies ($>50$ cells per colony) were counted using MetaMorph software (Molecular Devices).

**Apoptosis assay.** Cells were collected from 24-well plates, spun down and incubated in darkness at $4\,^{\circ}C$ with 0.5% annexin-V-APC antibody (BD Pharmigen) and $100 \mu g\,ml^{-1}$ propidium iodide dissolved in annexin-V binding buffer (1 M CaCl, 5 M NaCl, 1 M HEPES). At least $1 \times 10^4$ events were recorded by FACS (BD FACSCanto II, BD Bioscience) and analysed using Flowlogic software (Inivai Technologies).

**Cytochrome *c* release assay.** Cytochrome *c* release from the mitochondria was assayed as previously described[41]. Briefly, all cells were collected from 24-well plates, washed with PBS and permeabilized with 0.05% digitonin (Sigma-Aldrich, $10 \mu g\,ml^{-1}$ in PBS with 100 mM KCl) until $>95\%$ of cells were as determined by trypan blue exclusion. Cells were fixed immediately in 4% paraformaldehyde for 20 min at room temperature, washed thrice with PBS, incubated for 1 h in blocking buffer (3% BSA, 0.5% saponin in PBS) and probed overnight at $4\,^{\circ}C$ with anti-cytochrome *c* antibody (BD Pharmigen 556432, 1:200 in blocking buffer). Subsequently, cells were washed twice with PBS, incubated at room temperature for 1 h with AF-488 anti-mouse secondary antibody (ThermoFisher Scientific, 1:200 in blocking buffer) and analysed by FACS (BD FACSCanto II).

**ROS detection.** Mitochondrial-specific and generalized cellular ROS were detected using MitoSOX Red and CellROX Deep Red reagents respectively (ThermoFisher Scientific). ROS sensors ($5 \mu M$) in phenol-red free media were applied to cells in 96-well plates for the entire study duration. ROS-induced fluorescence was quantitated using a Cellomics ArrayScan VTI HCS reader (ThermoFisher Scientific) and normalized to cell number.

**Lipid peroxidation detection.** Lipid peroxidation was detected using C11-BOD-IPY(581/591) dye (ThermoFisher Scientific). Cells were incubated with C11-BODIPY ($5 \mu M$) in 24-well plates for defined periods. All cells were collected, spun down, washed with PBS, resuspended in phenol-red free medium, and analysed using FACS (BD LSRFortessa BD Bioscience). At least $1 \times 10^4$ events were recorded.

**Intracellular glutathione assay.** Total intracellular GSH from tissues or cells (10 cm dishes) were assayed using the Cayman Chemicals Glutathione Kit as per the manufacturer's instructions. GSH concentration was calculated from an internal standard curve and normalized to tissue weights or total cell number as determined from parallel plates.

**Glutamate release assay.** System $x_c^-$ activity was assessed by measuring extracellular release of glutamate from H1299 cells. Overall, 20,000 cells were seeded into 96-well plates. The next day, cells were washed with PBS and incubated for 3 h in $50 \mu l$ of choline buffered solution (137.5 mM choline chloride, 5.36 mM KCl,

0.77 mM $KH_2PO_4$, 0.71 mM $MgSO_4.7H_2O$, 1.1 mM $CaCl_2$, 10 mM D-glucose and 10 mM HEPES, pH 7–7.5 at $37\,^{\circ}C$) containing $600 \mu M$ L-cystine $\pm 40 \mu M$ erastin or 1 mM SAS. Extracellular glutamate was detected using the Amplex Red glutamate release assay kit (Molecular Probes) as per the manufacturer's instructions. Of note: as the glutamate released by system $x_c^-$ is $Na^+$-independent, $Na^+$ was substituted with choline in the reaction buffer to increase the specificity of this assay for measuring system $x_c^-$ activity[42].

**Protein immunoprecipitation.** Four million H1299 mut-p53$^{R273H}$ cells or $5 \times 10^6$ FLO-1, Eso26 and OACM5.1 cells were seeded in 15 cm dishes. The next day, cells were collected in ice cold PBS, pelleted and sonicated in NP-40 lysis buffer (50 mM Tris pH 8, 150 mM NaCl, 5 mM EDTA, 0.5% NP-40) with protease inhibitors (Roche). An aliquot of lysate was removed for input sample prior to incubation with Protein A-Sepharose beads (Invitrogen). Immunoprecipitations were performed with anti-NRF2 antibody (Cell Signaling Technology, clone D1Z9C), anti-p53 antibody-agarose conjugate (Santa Cruz Biotechnology), IgG (Santa Cruz Biotechnology) or Protein A-Sepharose only. Beads were washed five times in NP-40 lysis buffer and once in lysis buffer without NP-40 then boiled in $2.5 \times$ sample buffer (125 mM Tris-HCl pH 6.8, 5% SDS, 25% glycerol, 2.5% 2-mercaptoethanol, 0.005% bromophenol blue). Samples were analysed by western blotting with anti-NRF2 (clone D1Z9C) and anti-p53 (clone DO-1) antibodies.

**Chromatin immunoprecipitation.** H1299 mut-p53$^{R273H}$ cells were seeded as per protein immunoprecipitation and treated the next day with $50 \mu M$ $H_2O_2$ for 4 h. Protein–DNA conjugates were cross-linked by the addition of 1% formaldehyde for 10 min. Cells were collected in PBS and sequentially lysed in buffer 1 (50 mM HEPES pH 7.5, 140 mM NaCl, 1 mM EDTA, 10% glycerol, 0.5% NP-40, 0.25% TritonX-100), buffer 2 (10 mM Tris pH 8, 200 mM NaCl, 1 mM EDTA, 0.5 mM EGTA) and buffer 3 (10 mM Tris pH 8, 100 mM NaCl, 1 mM EDTA, 0.5 mM EGTA, 0.1% sodium deoxycholate, 0.5% N-lauroylsarcosine) with protease inhibitors (Roche). Lysates were sonicated followed by precipitation of cellular debris with 1% Triton X-100. An aliquot of lysate was removed for input sample prior to immunoprecipitation as described above. Beads were washed five times in LiCl wash buffer (50 mM HEPES pH 7.5, 500 mM LiCl, 1 mM EDTA, 1% NP-40, 0.7% sodium deoxycholate), once in TE/NaCl wash buffer (10 mM Tris pH 8, 1 mM EDTA, 50 mM NaCl) and protein–DNA conjugates eluted by incubating at $65\,^{\circ}C$ in 50 mM Tris pH 8 with 10 mM EDTA and 1% SDS, followed by a further 6 h incubation at $65\,^{\circ}C$. DNA was recovered by phenol:chloroform:isoamyl alcohol (25:24:1, Sigma-Aldrich) and ethanol/sodium acetate precipitation following digestion with $0.2 mg\,ml^{-1}$ RNAseA (Promega) and $0.2 mg\,ml^{-1}$ Proteinase K (Promega). Samples were analysed by RT-PCR using primers bordering a cluster of three NRF2 consensus sites in the *SLC7A11* promoter (forward 5′–3′: AGGCTTCTCATGTGGCTGAT; reverse 5′–3′: AGAATTGAGAGCACGA TGCA).

**High performance liquid chromatography and mass spectrometry.** Synthetic GS-MQ or APR-246 treated de-proteinized JH-EsoAd1 cell lysates were resolved on an ODS-Hypersil C18 HPLC column ($2.1 \times 100$ mm, $5 \mu m$, Hewlett-Packard) in 10 mM $NH_4HCO_3$ pH 8, using a 0–80% acetonitrile gradient over 10 min. Flow rate was $0.1 ml\,min^{-1}$. Fractions of $25 \mu l$ were collected and acidified with 2% formic acid for mass spectrometric analysis. All analyses were done on an Agilent 6220 ESI-TOF mass spectrometer. Mass spectrometry data was acquired and reference mass corrected via a dual-spray electrospray ionization (ESI) source. Mass spectra were created by averaging the scans across each peak and background subtracted against the first 10 s of the total ion current. Acquisition was performed using the Agilent Mass Hunter Acquisition software version B.02.01 (B2116.30) and analysed using Mass Hunter version B.03.01. GS-MQ was detected at a 445.17 *m/z* peak.

**Light microscopy.** Phase contrast images were acquired using an AMG EVOS FL (Advanced Microscopy Group) microscope. Four independent fields were taken per experimental condition. Representative images are shown.

**Transmission electron microscopy.** FLO-1 and JH-EsoAd1 cells were seeded in 10 cm dishes. The following day, cells were treated with vehicle ($H_2O$) or APR-246 (FLO-1: $25 \mu M$ and JH-EsoAd1: $40 \mu M$) for 15 and 24 h. Cells were collected in PBS, gently spun down, fixed in 2% paraformaldehyde, 2.5% glutaraldehyde in 0.1 M sodium cacodylate buffer before washing in 0.1 M sodium cacodylate buffer and post-fixing in 1% osmium tetroxide in 0.1 M sodium cacodylate buffer. Following fixation, cells were dehydrated through a graded series of alcohols and embedded in Spurrs Resin. Ultrathin sections were cut with a diamond knife using a Leica Ultracut S ultra-microtome (Austria), stained with both methanolic uranyl acetate and lead citrate before viewing in a JEOL 1011 transmission electron microscope (Japan) at 60 kV. Images were recorded with a MegaView III CCD cooled digital camera (Soft Imaging Systems, Germany) at $\times 10,000$–$\times 80,000$ magnification. At least 10 mitochondria were examined per treatment group. Representative images are shown.

**Combination drug studies.** To quantify the synergistic activity of APR-246 with system $x_C^-$ inhibitors, the GI50 dose of single agents was firstly determined by fitting the Hill equation using Prism 6 software (Graphpad). Cells were then treated for 96 h with combinations of APR-246 and system $x_C^-$ inhibitors over a range of concentrations held at a fixed ratio and based on the GI50 of each drug. The highest and lowest combination ratios were three times and 1/10th the GI50, respectively. Combination indexes (CI) at 50%, 75% and 90% reduction in cell viability was determined using CalcuSyn v2 (Biosoft) where synergism: CI < 0.9, antagonism: CI > 1.1 and additive effect: 0.9 ≤ CI ≤ 1.1 (ref. 43).

**Xenograft models and treatment.** All animal experiments were approved by the Peter MacCallum Cancer Centre (PMCC) Animal Experimentation Ethics Committee and undertaken in accordance with the National Health and Medical Research Council Australian Code of Practice for the Care and Use of Animals for Scientific Purposes. For cell line xenografts, 5 million cells suspended in 100 µl of 1:1 PBS and Matrigel (BD Bioscience) were subcutaneously injected into the flank of 6 weeks old, female nude or NOD-SCID IL-2Rγ$^{KO}$ (NSG) mice. Patient-derived xenografts (PDX) were established as previously described and implanted subcutaneously on the dorsum of NSG mice[44]. The collection and use of human tissue was approved by the Human Research Ethics Committee at PMCC. Informed consent was received from all patients. Tumour volume was assessed unblinded with caliper measurements every 3–4 days, and calculated using the formula (length × width$^2$)/2. Mice were randomized to treatment cohorts once tumours reached 80 mm$^3$. For all experiments, APR-246 was administered at 100 mg kg$^{-1}$ daily and SAS at 6 mg per mouse twice daily. Saline was given to control animals. All treatments were delivered by intraperitoneal injections. For shRNA experiments, sh*SLC7A11* was induced with doxycycline chow (600 mg kg$^{-1}$, Specialty Feeds, Australia) and water (2 mg ml$^{-1}$, Sigma-Aldrich) available *ad libitum*. Three days following doxycycline induction, mice were imaged using the Maestro2 (CRI) fluorescence imaging system to detect GFP expression. Tumours were subsequently monitored as above. All mice were killed when tumours reached 1,500 mm$^3$ for survival studies or 6 h after the last treatment dose for biomarker studies. Tumours were photographed, weighed and partitioned for GSH, immunohistochemical, RT-PCR and western blot analysis. Tumour growth inhibition was calculated with the formula [1-(Tf-Ti)/mean(Cf-Ci)] × 100, where Tf, Ti and Cf, Ci represents final (f) and initial (i) tumour volume of drug treated (T) and control (C) animals respectively. On the basis of the sample size calculations, n = 5 per group will detect a 50% difference with a power of 80% and an alpha error of 5%.

**Immunohistochemistry.** Sections from formalin-fixed paraffin embedded tissues were stained with Ki67 (ab16667, Abcam) and anti-cleaved caspase-3 (9664, Cell Signalling Technology) antibodies, detected using the Envision Plus system (Dako), and viewed on a BX51 microscope (Olympus). Three representative images per tumour were captured at 20 × magnification and analysed using MetaMorph software.

**TCGA data analysis.** The cBioportal for Cancer Genomics (www.cbioportal.org) was used to analyse the relationship between p53 protein expression and *SLC7A11* or *NQO1* mRNA expression across all available RPPA and RNAseq datasets. RNAseq data was linearized, and a protein z-score ≥ 1 was defined as high p53 expression.

**NCI-60 pharmacogenomic analysis.** The Pattern comparison function from CellMiner 1.6 (http://discover.nci.nih.gov/cellminer/analysis.do) was used to identify novel predictors of PRIMA-1 (NSC 281668) sensitivity across the NCI-60 cancer cell line panel[13]. The mean-normalized transcript intensity z-scores from 5 Agilent expression arrays were correlated with PRIMA-1 GI50 (µM following antilog transformation). Genes with a |r| > 0.3 are considered by the National Cancer Institute's Therapeutics Development programme to be potentially significant predictors of drug sensitivity[13], and thus were shortlisted for further analysis.

**Project Achilles multi-omics data analysis.** Project Achilles v2.4 (www.broadinstitute.org/achilles) is a genome-wide shRNA screen performed in 216 cell lines. Cells were transduced with five shRNAs per gene. The abundance of remaining individual shRNAs relative to an initial reference pool was quantified by next-generation sequencing, and reflects each cell line's sensitivity to knockdown of that gene[16]. Gene-level shRNA scores were derived by calculating the median from individual shRNA scores. A gene-level shRNA score below 0 indicates reduced cell viability, equal to 0 indicates no effect on viability, and above 0 indicates enhanced viability. The *TP53* status for each cell line was annotated from the Cancer Cell Line Encyclopaedia (www.broadinstitute.org/ccle) and the IARC *TP53* Database (p53.iarc.fr) (Supplementary Fig. 9). Fifteen cell lines were excluded from the final analysis due to unknown or conflicting *TP53* status. In addition, a comprehensive list of 548 antioxidative stress genes were compiled from several sources: (1) Chromatin immunoprecipitation sequencing of NRF2 under basal and ROS-induced conditions as identified by Chorley *et al.*[27], (2) known human genes with functionally validated antioxidant responsive elements as reviewed by Chorley

*et al.*[27] and (3) Gene Ontology designated ROS responsive genes[45]. These antioxidative stress genes were interrogated using Project Achilles v2.4. to identify candidate genes, which when inhibited would preferentially reduce the viability of missense mut-p53 cancer cells (which typically accumulate mut-p53 protein) compared with cells that are p53 wild-type or null (non-missense p53, which typically have low or absent p53 levels). The extent of preferential killing was calculated by subtracting the median gene-level shRNA score of missense mut-p53 cell lines from the median gene-level shRNA score of non-missense p53 cell lines, such that a value above 0 indicates preferential killing of missense mut-p53 cells, and a value below 0 indicates preferential killing of non-missense p53 cells.

**Statistics.** Data were analysed with Student's *t*-test or ANOVA with Tukey's/Dunnett's multiple comparison post-test. Survival differences were compared using Kaplan–Meier log-rank analysis. Correlation between two groups was evaluated by the Pearson's test. Statistical analyses were performed using Prism 6 (Graphpad) with P < 0.05 considered statistically significant.

**Data availability.** The datasets analysed within this study are available from cBioportal for Cancer Genomics (www.cbioportal.org), CellMiner 1.6 (http://discover.nci.nih.gov/cellminer/analysis.do), Project Achilles v2.4 (www.broadinstitute.org/achilles), Cancer Cell Line Encyclopedia (www.broadinstitute.org/ccle), IARC TP53 Database (p53.iarc.fr), Nucleic Acids Research online (DOI: https://doi.org/10.1093/nar/gks409) and Oncotarget online (DOI: 10.18632/oncotarget.1658). The authors declare that all other data supporting the findings of this study are available within this paper and its supplementary information files.

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

## Acknowledgements

This work was supported by a National Health and Medical Research Council (NHMRC) of Australia Centres for Research Excellence Grant #1040947 (W.A.P.), a Translational Research Project Grant (TRP15012) from the Victorian Government Department of Health and Human Services through the Victorian Cancer Agency (W.A.P., N.J.C., C.P.D.), and a Peter MacCallum Cancer Foundation New Investigator grant (D.S.L.); D.S.L. was supported by Royal Australasian College of Surgeons (RACS) Foundation for Surgery John Loewenthal, Reg Worcester and Eric Bishop Fellowships, Cancer Therapeutics Scholarship (Cancer Therapeutics CRC) and NHMRC Postgraduate Research Scholarship; O.M.F. by a NHMRC Postgraduate Research Scholarship and a Swiss National Science Foundation Doc.Mobility Grant; M.R. by a RACS Thornell-Shore Memorial Scholarship and Melbourne University Sir Thomas Naghten Fitzgerald Scholarship; and G.R.G. by a RACS Foundation for Surgery Research Scholarship and a Covidien Colorectal Research Fellowship. N.J.C. is supported by a Victorian Cancer Agency Fellowship (MCRF16002) and an NHMRC Project Grant #1120293 (W.A.P., N.J.C., Y.H.). H.B.P. is supported by a Marie-Sklodowska Curie Actions/Sêr Cymru COFUND fellowship. We thank Prof Johannes Zuber for providing the tet-inducible shRNA vector and Prof Ricky Johnstone, Prof Mark Dawson, Dr Daniella Brasacchio, Dr Liz Christie, and the Peter MacCallum Cancer Centre FACS, microscopy, functional genomics and animal core facilities, for their advice and/or technical assistance.

## Author contributions

D.S.L., C.M.H., S.H., Y.H., C.P.D., N.J.C. and W.A.P. were responsible for overall study concept and design of experiments. D.S.L., N.J.C., K.G.M., C.M.H., H.B.P., O.M.F., M.R. and G.R.G. acquired and analysed data. K.G.M., C.M.H., W.J.A., H.B.P., O.M.F., M.R., S.H., C.C., K.G.W. and L.A. provided technical and material support. D.S.L., C.P.D., N.J.C. and W.A.P. obtained funding for this study. All were responsible for interpretation of data and drafting of manuscript.

## Additional information

**Competing interests:** K.G.W. is a cofounder and shareholder of Aprea Therapeutics, and L.A. is the chief scientific officer for Aprea Therapeutics, a company that develops p53-based anticancer therapy including APR-246. All other authors declare no competing financial interests.

