## [Peer Review File · Nature Communications]

Reviewers' comments:

Reviewer #1 (Remarks to the Author):

This is an exciting and important paper that discovers the mechanism of action of the small molecule drug APR246, which is currently showing great promise in the clinic in the treatment of cancer. The discovery of the mechanism of action suggests clear criteria for patient selection and rational drug combinations and the work is thus of great and immediate translational value. At a more basic level the study provides a clear proof that the expression of high levels of mutant p53 protein in a cancer creates a specific vulnerability to reactive oxygen species. This is the mutant p53 protein function to inhibit the NRF2 transcription factor by binding to it and blocking its ability to drive the anti-oxidant response. The study is very comprehensive and is suitable for publication.

The paper is very well written and helps to explain and clarify a large earlier literature on the drug APR 246 which was initially discovered in a screen precisely developed to find molecules that synergized with mutant p53 expression to kill cells. APR 246 is a very small molecule and reacts with cys groups on proteins. In initial studies, since confirmed by many, it was found that the drug could modify mutant p53 covalently and that this could restore the DNA binding, wild type conformation and transcriptional activity of mutant p53. These remarkable findings became harder to interpret when it became clear that APR 246 could modify many proteins with reactive cys and could be toxic to cells that did not express mutant p53.

In this paper the authors show that the basis of the mutant p53 selectivity of APR 246 is more subtle and that it in fact depends on the molecules ability to inactivate glutathione. In cells where glutathione levels are low due to the newly discovered function of mutant p53 this results in cell death. Remarkably then the reactivation of mutant p53 is not relevant and would in fact be expected to be slightly detrimental to the action of the drug.

The whole data and model are very nicely summarized in figure 9 and I do not intend here to go through very figure.

The authors start by showing that APR 246 is toxic to mutant p53 expressing cells through ROS, as killing is blocked by antioxidants such as N-acetyl cysteine and GSH monethyl ester. They go on to confirm the drugs reaction with and neutralization of GSH. They then use bioinformatics data to show a remarkable link between cells sensitivity to APR 246 and the levels of the mRNA for SLC7A11. This turns out to be the clue to the mystery of p53 mutant cells being more sensitive to APR246. Firstly using SLC7A11si RNA knock down and SLC7A11 over expression they confirm the functional link between SLC7A11 and sensitivity to APR 246. Since SLC7A11 is a key functional component of the cystine/glutamate antiporter system xc they anticipate, and then prove, in cells and in animals that APR 246 synergizes strongly with system Xc inhibitors such as Erastin and SAS.

The final part of the puzzle is the link between low levels of SLC7A11 mRNA and high levels of mutant p53. Here the authors show that this is because SLC7A11 mRNA is induced by the NRF2 transcription factor whose activity is inhibited by binding to mutant p53. The interaction between NRF2 and mutant p53 has been recently reported by others but this does not detract from the novelty of this manuscript as the focus here is on the complex story of APR 246

I think the paper will provoke great interest in a wide audience. It is a great example of using every modern tool to determine drug action and offers real hope for p53 directed cancer therapy, which has been a holy grail for many.

The figure quality is high but I had some difficulty with the colors used in the graphs most notable in fig 7 a where some of the data lines are hard to see.

David Lane

Reviewer #2 (Remarks to the Author):

1. Please comment on your findings in the context of the research recently published by Jiang et al 2015 in Nature, especially with regard to their report that wild-type p53 suppresses SLC7A11 expression, is able to bind to a p53 consensus sequence in the SLC7A11 promoter region, and that its knockdown abrogates SLC7A11 down-regulation.

2. The authors suggest in Figure 6B that levels of SLC7A11 and the p53 status may be correlated across different panels of cancer cells, including breast cancers. We would like to point out a paper by Timmerman et al 2013, which has examined a large panel of breast cancer cells in detail with regard to glutamine consumption, system xc⁻ activity, and SLC7A11 expression. We believe that it is important to examine not only the expression of SLC7A11, but to also include a measure of its function, either via glutamate release or cystine uptake/consumption, as a key measure when assessing the correlation of p53 mutation status and SLC7A11. This is important given that, according to Timmerman's paper and the p53 status of various breast cancer cells (Neve et al 2006), the following can be observed: MDA-MB-231 cells (mutated p53) have high SLC7A11 expression and high cystine consumption/glutamate release, while BT549 cells (mutated p53) have high SLC7A11 expression and low cystine consumption/glutamate release. In addition, in our lab, we have shown that T47D cells (mutated p53) express significantly lower levels of SLC7A11 protein (Linher-Melville et al 2016), also releasing significantly less glutamate/taking up less cystine. Interestingly, Timmerman et al have included the non-tumorigenic breast epithelial cell line 184B5 in their analysis, which is derived from a young woman undergoing breast reduction. This cell line displays high SLC7A11 expression, but very low low cystine consumption/glutamate release.

We would like the authors to comment on these correlations due to their widening the scope of the research findings to cancers other than esophageal, and to also include either glutamate release or cystine uptake assays to examine the functional relevance of system xc⁻ when correlating p53 status to SLC7A11 expression. GSH and ROS can be affected by changes in numerous cellular mechanisms, and it is therefore important to address this point.

3. In line 25, page 7, as well as in the discussion, lines 6-7, page 11, the authors suggest that erastin is a specific system xc⁻ inhibitor. We agree, from first-hand experience, that SAS rapidly (within 10 min) and dramatically (by 80%) blocks cystine uptake (and glutamate release), demonstrating a specific direct effect on this transporter. However, we are not aware of a similar direct effect for erastin. Has this been clearly established? Otherwise, it is working through off-target effects, as it is known to have an effect on mitochondrial function by producing non-apoptotic cell death by affecting voltage-dependent anion channel gating, allowing cations to enter mitochondria that culminates in the release of oxidative species and oxidative cell death (Yagoda et al 2007, Nature). The authors' data and information in the discussion on erastin should be addressed in the context of this information –it should be presented as an ROS-inducer rather than a system xc⁻ inhibitor.

In lines 3-6, page 7, the authors discuss that mut-p53 is associated with high ROS levels and low GSH reserves. If SLC7A11 is also low in these cells, how are they coping with oxidative stress to continue to survive? What other mechanisms are at play? Please discuss.

With regard to combination therapies, you have shown that blocking system xc⁻ activity together with APR-246 works to deplete intracellular GSH levels. However, this is a short-term effect that relies on continuous readministration of system xc⁻ blockers (if they are acting stearically). Please comment on the long-term effects of APR-246: SLC7A11 expression would rise in these cells. In the absence of a means to block system xc⁻ activity, would cells be more aggressive, given that SLC7A11 is emerging as a marker of "aggressive" cancers?

Other points to address:

In Figure 4a and in all other relevant figures, it should be clearly indicated that mut-p53, not its wild-type form, is being detected by Western blot analysis.

In Figure 4f, why is there no input band for NRF2 in the Eso26 IP?

Could you please show a larger area of your NRF2 IPs in Figure 4f? Given that there are very few "clean" NRF2 antibodies that are commercially available that detect only the expected band for this protein on an immunoblot (please see the paper by Kemmerer et al 2015), it would be encouraging to see more of the blot to verify specificity of the interaction.

Line 31, page 6: "...NRF2-mediated transcriptional up-regulation of SLC7A11 (Fig. 4h)." The authors assessed mRNA levels of SLC7A11, but did not carry out any specific transcriptional assays (such as promoter studies).

Reviewers' comments:

Reviewer #1 (Remarks to the Author):

The figure quality is high but I had some difficulty with the colors used in the graphs most notable in fig 7a where some of the data lines are hard to see.

We sincerely thank Professor David Lane for his comprehensive, insightful, and very positive feedback on our manuscript.

As suggested, we have adjusted the color pallet in Fig 7a to make the data lines more visible.

Reviewer #2 (Remarks to the Author):

1. Please comment on your findings in the context of the research recently published by Jiang et al 2015 in Nature, especially with regard to their report that wild-type p53 suppresses SLC7A11 expression, is able to bind to a p53 consensus sequence in the SLC7A11 promoter region, and that its knockdown abrogates SLC7A11 down-regulation.

In the publication by Jiang et al (1), the authors demonstrated that increased wt-p53 protein resulted in decreased SLC7A11 expression. This was postulated to result from trans-repression of the SLC7A11 gene following wt-p53 binding to the SLC7A11 promoter. These authors however, did not rule out the possibility that suppression of SLC7A11 by wt-p53 may also involve NRF2. Indeed, the consensus sequence for NRF2 on the SLC7A11 promoter is immediately adjacent to that of wt-p53 (see Fig. A below). Although this was not described by Jiang et al, it suggests that wt-p53 may interact with NRF2 in the regulation of SLC7A11 expression. Nonetheless, we have commented on this in our discussion section by adding the following:

"...a mechanism which is separate from the transcriptional regulation of SLC7A11 by wt-p53 described by Jiang et al.¹⁷"

```
5'.....TGCTGGAGGCTTCTCATGTGGCTGATGCAAACCTGGAGAATTGCATCATCATTTAGCTGTA  
GTAAGTTGGTGTGACAGGCAGGCGCTTAAATACAAGCCCATGAGGAAGCTGAGCTGGTTTGTAA  
TGATAGGGCGGCAGCAGCAGCAGCAGCAGCAGTGGTGGAAACGAGGAGGTGGAGAATTGAGAG  
CACGATGCATACACAGGTGTTTCTGAGTAGTAATTAGATCGCTGTGAAGGAAAAAGCACACCTTT  
GAGTTTTACCTGTGAACACTATAGCGCTGAGAGAGACAGTCTGAAAGCAGAGGAAGACATCGA  
TCAGTAACACCAAGAGACACCAAAGTTGAAAGTTTTGTTTTCTTCCCTCTGTTTTATTTTTCCCC  
GTGTGCCCTACTATG.....3'
```

Fig A. Wt-p53 and NRF2 consensus binding sites on the SLC7A11 promoter region. Shown is a 400bp DNA sequence of the promoter region, 5' UTR and transcriptional start site of the human SLC7A11 gene. ATG refers to the transcriptional start site. Letters in red refer to the 5' UTR. Letters in green refer to part of the promoter sequence. Highlighted in black is the wt-p53 binding location as per Jiang et al (1). Highlighted in blue is the NRF2 binding location as per Chorley et al (2). The close proximity of wt-p53 and NRF2 binding sites potentially suggest p53-NRF2 interaction in the transcriptional regulation of SLC7A11.

2. The authors suggest in Figure 6B that levels of SLC7A11 and the p53 status may be correlated across different panels of cancer cells, including breast cancers. We would like to point out a paper by Timmerman et al 2013, which has examined a large panel of breast cancer cells in detail with regard to glutamine consumption, system xc- activity, and SLC7A11 expression. We believe that it is important to examine not only the expression of SLC7A11, but to also include a measure of its function, either via glutamate release or cystine uptake/consumption, as a key measure when assessing the correlation of p53 mutation status and SLC7A11. This is important given that, according to Timmerman's paper and the p53 status of various breast cancer cells (Neve et al 2006), the following can be observed: MDA-MB-231 cells (mutated p53) have high SLC7A11 expression and high cystine consumption/glutamate release, while BT549 cells (mutated p53) have high SLC7A11 expression and low cystine consumption/glutamate release. In addition, in our lab, we have shown that T47D cells (mutated p53) express significantly lower levels of SLC7A11 protein (Linher-Melville et al 2016), also releasing significantly less glutamate/taking up less cystine. Interestingly, Timmerman et al have included the non-

tumorigenic breast epithelial cell line 184B5 in their analysis, which is derived from a young woman undergoing breast reduction. This cell line displays high SLC7A11 expression, but very low cystine consumption/glutamate release. We would like the authors to comment on these correlations due to their widening the scope of the research findings to cancers other than esophageal, and to also include either glutamate release or cystine uptake assays to examine the functional relevance of system xc⁻ when correlating p53 status to SLC7A11 expression. GSH and ROS can be affected by changes in numerous cellular mechanisms, and it is therefore important to address this point.

We thank the reviewer for their detailed commentary. While we agree that there are examples of cell lines where mut-p53 status does not correlate with SLC7A11 expression, we believe that there are several reasons for this. Firstly, as emphasized throughout the manuscript, it is the level of mut-p53 protein and not the p53 mutation status per se that correlates with SLC7A11 expression. Since mutation of TP53 does not necessarily equate to accumulation of mut-p53 protein, being categorized as "mutated p53" may not necessarily correlate with SLC7A11 expression. Secondly, as described in the discussion section, SLC7A11 expression may also be regulated by other proteins including c-Myc, CD44v and ATF4. Thirdly, some cancer cells may have SLC7A11 gene copy amplification (<http://cansar.icr.ac.uk/>). These factors may modulate the relationship between mut-p53 and SLC7A11.

Nevertheless, we agree with the reviewer that assessing system xc⁻ activity is important. To this end, we have used a glutamate release assay to compare the activity of system xc⁻ in H1299 cells that are p53^{Null} versus those expressing p53^{R273H} and p53^{R175H} mutants. In this system, mut-p53 overexpression resulted in reduced levels of SLC7A11 and was associated with reduced system xc⁻ activity. These results are now presented in **Fig. 5c** of the revised manuscript. The associated text has also been updated accordingly.

- 3. In line 25, page 7, as well as in the discussion, lines 6-7, page 11, the authors suggest that erastin is a specific system xc⁻ inhibitor. We agree, from first-hand experience, that SAS rapidly (within 10 min) and dramatically (by 80%) blocks cystine uptake (and glutamate release), demonstrating a specific direct effect on this transporter. However, we are not aware of a similar direct effect for erastin. Has this been clearly established? Otherwise, it is working through off-target effects, as it is known to have an effect on mitochondrial function by producing non-apoptotic cell death by affecting voltage-dependent anion channel gating, allowing cations to enter mitochondria that culminates in the release of oxidative species and oxidative cell death (Yagoda et al 2007, Nature). The authors' data and information in the discussion on erastin should be addressed in the context of this information –it should be presented as an ROS-inducer rather than a system xc⁻ inhibitor.**

Two major publications by Dixon et al have demonstrated that erastin does in fact inhibit system xc⁻ activity (3, 4). Moreover, we now have our own data showing that erastin reduces glutamate release by system xc⁻. This data is now presented in **Fig. S6c** of the revised manuscript. The associated text has also been updated accordingly.

- 4. In lines 3-6, page 7, the authors discuss that mut-p53 is associated with high ROS levels and low GSH reserves. If SLC7A11 is also low in these cells, how are they coping with oxidative stress to continue to survive? What other mechanisms are at play? Please discuss.**

As demonstrated in Fig 4 and 5 of the manuscript, mut-p53 mediated suppression of *SLC7A11* expression results in higher basal ROS levels and lower GSH reserves. This increase in ROS levels on its own is insufficient to induce cell death, as there are multiple pathways and molecules inside the cell that help defend against low levels of oxidative stress (5). The fact that these cells have lost wt-p53 function (impaired DNA damage response, with limited ability to undergo cell cycle arrest and apoptosis) likely enhances their ability to survive in such circumstances (6). Furthermore, heightened ROS levels have been associated with oncogenic signalling and tumorigenesis in esophageal adenocarcinoma and other cancer types (7-9).

As highlighted in our results and discussion section, the key point is that cancer cells with accumulated mut-p53 protein, compared to normal cells, are more susceptible to further oxidative stress. This creates a therapeutic window which enables preferential killing of tumor cells that accumulate mut-p53 protein. To this end, our study demonstrates that these cells are particularly vulnerable to inhibition of system x_c^- . That is, in cancer cells with accumulation of mut-p53 protein, low level system x_c^- blockade, which minimally affects normal cells, further increases ROS, pushing levels across an already compromised threshold, resulting in their demise.

5. With regard to combination therapies, you have shown that blocking system x_c^- activity together with APR-246 works to deplete intracellular GSH levels. However, this is a short-term effect that relies on continuous re-administration of system x_c^- blockers (if they are acting stearically). Please comment on the long-term effects of APR-246: *SLC7A11* expression would rise in these cells. In the absence of a means to block system x_c^- activity, would cells be more aggressive, given that *SLC7A11* is emerging as a marker of “aggressive” cancers?

We agree with the reviewer's hypothesis that long term treatment with APR-246 would likely induce *SLC7A11* expression, potentially leading to resistance. We propose that combining APR-246 with system x_c^- blockade will not only lead to synergistic induction of tumor cell death (as shown in Fig 7 and 8 of the manuscript), but may also inhibit the development of cellular resistance to APR-246, thus further illustrating the importance of our study.

It is harder to predict whether the increase in *SLC7A11* expression subsequent to APR-246 treatment would render tumors more "aggressive". While there are studies which have correlated high *SLC7A11* expression with advanced disease in some cancers, in our view there is insufficient evidence to demonstrate that *SLC7A11* itself drives tumor progression/aggressiveness. Moreover, phase Ib data from the EUTROC PiSARRO trial, which evaluated APR-246 in combination with conventional chemotherapy for high grade serous ovarian cancer, achieved a response rate of 78%, and disease control in 100% of patients who had relapsed following first line chemotherapy (10). These results would suggest that repeated treatment with APR-246 in patients does not increase tumor aggressiveness.

6. In Figure 4a and in all other relevant figures, it should be clearly indicated that mut-p53, not its wild-type form, is being detected by Western blot analysis.

These figures have been amended as per the reviewer's suggestion.

7. In Figure 4f, why is there no input band for NRF2 in the Eso26 IP?

Since the input sample only represented 4% of the total cell lysate, whilst each immunoprecipitation sample was performed on 24% of the total cell lysate, it is to be anticipated that the NRF2 band would be significantly less intense in the input lane compared to the IP lane. Furthermore, to avoid overexposure of the IP lanes, we presented an immunoblot with shorter exposure time, thus the input band appears faint. Below, we provide the same blot captured after prolonged exposure. As can be seen, there is clearly an input band for NRF2 (**Fig B**).

Fig B. Immunoblot of NRF2 following immunoprecipitation of NRF2 and p53 from Eso26 cell lysates. The upper and lower blots were exposed for 2 and 5 minutes respectively. Whilst the NRF2 band is faintly visible in the input column from the upper blot, this can be clearly seen in the lower blot.

8. Could you please show a larger area of your NRF2 IPs in Figure 4f? Given that there are very few “clean” NRF2 antibodies that are commercially available that detect only the expected band for this protein on an immunoblot (please see the paper by Kemmerer et al 2015), it would be encouraging to see more of the blot to verify specificity of the interaction.

As requested by the reviewer, the extended immunoblots for each immunoprecipitation experiment are shown below. The red dotted box indicates the area cropped in Fig. 4f. The blue dotted box indicates the area cropped for the reverse IPs in Fig. S4h. In addition, we have also provided an immunoblot of NRF2 following genetic knockdown of NRF2 in Eso26 cells to demonstrate the specificity of the antibody. These blots have been included into the supplementary section of the revised manuscript.

Eso26 cells were treated with NRF2 or non-targeting control (NTC) siRNA for 72 hr prior to western blot.

- 9. Line 31, page 6: "...NRF2-mediated transcriptional up-regulation of SLC7A11 (Fig. 4h)." The authors assessed mRNA levels of SLC7A11, but did not carry out any specific transcriptional assays (such as promoter studies).**

Given that we did not carry out any promoter studies to demonstrate that NRF2 transcriptionally upregulates *SLC7A11*, we agree with the reviewer that the word "transcriptional" should be removed from line 31, page 6. This sentence now reads "*...NRF2-mediated upregulation of SLC7A11 (Fig. 4h)*".

Nonetheless, it should be noted that 1) *SLC7A11* is a well documented transcriptional target of NRF2 (2), 2) there are multiple high affinity binding sites for NRF2 in the promoter region of *SLC7A11* (as per Reviewer #2 comment 1 above), and 3) NRF2 knockdown in H1299 cells diminished *SLC7A11* mRNA levels (Fig. S4B).

References

1. Jiang L, Kon N, Li T, Wang SJ, Su T, Hibshoosh H, et al. Ferroptosis as a p53-mediated activity during tumour suppression. *Nature*. 2015;520:57-62.
2. Chorley BN, Campbell MR, Wang X, Karaca M, Sambandan D, Bangura F, et al. Identification of novel NRF2-regulated genes by CHIP-Seq: influence on retinoid X receptor alpha. *Nucleic Acids Res*. 2012;40:7416-29.
3. Dixon SJ, Lemberg KM, Lamprecht MR, Skouta R, Zaitsev EM, Gleason CE, et al. Ferroptosis: an iron-dependent form of nonapoptotic cell death. *Cell*. 2012;149:1060-72.
4. Dixon SJ, Patel DN, Welsch M, Skouta R, Lee ED, Hayano M, et al. Pharmacological inhibition of cystine-glutamate exchange induces endoplasmic reticulum stress and ferroptosis. *eLife*. 2014;3:e02523.
5. Birben E, Sahiner UM, Sackesen C, Erzurum S, Kalayci O. Oxidative stress and antioxidant defense. *World Allergy Organ J*. 2012;5:9-19.
6. Kalo E, Kogan-Sakin I, Solomon H, Bar-Nathan E, Shay M, Shetzer Y, et al. Mutant p53R273H attenuates the expression of phase 2 detoxifying enzymes and promotes the survival of cells with high levels of reactive oxygen species. *J Cell Sci*. 2012;125:5578-86.
7. Nones K, Waddell N, Wayte N, Patch AM, Bailey P, Newell F, et al. Genomic catastrophes frequently arise in esophageal adenocarcinoma and drive tumorigenesis. *Nat Commun*. 2014;5:5224.
8. Mahalingaiah PK, Singh KP. Chronic oxidative stress increases growth and tumorigenic potential of MCF-7 breast cancer cells. *PloS One*. 2014;9:e87371.
9. Reuter S, Gupta SC, Chaturvedi MM, Aggarwal BB. Oxidative stress, inflammation, and cancer: how are they linked? *Free Radic Biol Med*. 2010;49:1603-16.
10. Basu B, Gourley C, Gabra H, Vergote IB, Brenton JD, Abrahmsen L, et al. PISARRO: A EUTROC phase 1b study of APR-246 with carboplatin (C) and pegylated liposomal doxorubicin (PLD) in relapsed platinum-sensitive high grade serous ovarian cancer (HGSOC). *Ann Oncol*. 2016;27:114-35. 10.1093/annonc/mdw368.

REVIEWERS' COMMENTS:

Reviewer #2 (Remarks to the Author):

The Authors have addressed most of my concerns. The only assay not conducted is the cysteine uptake but they show a glutamate release. This is acceptable but is an indirect measurement. I recommend its publication.

REVIEWERS' COMMENTS:

Reviewer #2 (Remarks to the Author):

The Authors have addressed most of my concerns. The only assay not conducted is the cysteine uptake but they show a glutamate release. This is acceptable but is an indirect measurement. I recommend its publication.

We are pleased the reviewer is happy that we have addressed their concerns and that they agree that, although indirect, our glutamate assay is an acceptable measure of system x_c^- activity in the context of this manuscript

We would like to sincerely thank reviewer #2 for their support in this manuscript's publication.